# Insulin and TOR signal in parallel through FOXO and S6K to promote epithelial wound healing

Parisa Kakanj[1,2,3], Bernard Moussian[4], Sebastian Grönke[2], Victor Bustos[2], Sabine A. Eming[3,5,6], Linda Partridge[2,6,7] & Maria Leptin[1,3,8]

The TOR and Insulin/IGF signalling (IIS) network controls growth, metabolism and ageing. Although reducing TOR or insulin signalling can be beneficial for ageing, it can be detrimental for wound healing, but the reasons for this difference are unknown. Here we show that IIS is activated in the cells surrounding an epidermal wound in *Drosophila melanogaster* larvae, resulting in PI3K activation and redistribution of the transcription factor FOXO. Insulin and TOR signalling are independently necessary for normal wound healing, with FOXO and S6K as their respective effectors. IIS is specifically required in cells surrounding the wound, and the effect is independent of glycogen metabolism. Insulin signalling is needed for the efficient assembly of an actomyosin cable around the wound, and constitutively active myosin II regulatory light chain suppresses the effects of reduced IIS. These findings may have implications for the role of insulin signalling and FOXO activation in diabetic wound healing.

[1] Institute for Genetics, University of Cologne, Zülpicherstr. 47a, Cologne 50674, Germany. [2] Max Planck Institute for Biology of Ageing, Joseph-Stelzmann-Str. 9b, Cologne 50931, Germany. [3] Center for Molecular Medicine Cologne, University of Cologne, Robert-Koch-Str. 21, Cologne 50931, Germany. [4] Institute of Biology Valrose (IBV), University of Nice-Sophia Antipolis, Nice Cedex 2 06108, France. [5] Department of Dermatology, University of Cologne, Kerpenerstr. 62, Cologne 50937, Germany. [6] Cologne Excellence Cluster on Cellular Stress Responses in Aging-Associated Diseases, University of Cologne, Joseph-Stelzmann-Str. 26, Cologne 50931, Germany. [7] Institute of Healthy Ageing, Department of Genetics, Evolution, and Environment, University College London, London WC1E 6BT, UK. [8] European Molecular Biology Laboratory, Meyerhofstr. 1, Heidelberg 69117, Germany. Correspondence and requests for materials should be addressed to L.P. (email: Partridge@age.mpg.de) or to M.L. (email: mleptin@uni-koeln.de).

The conserved IIS and TOR signalling network has a central role in metabolic homoeostasis, growth control, stress responses and ageing[1]. Decreased activity of the network can extend lifespan, improve many aspects of health and delay or prevent ageing-related diseases in laboratory animals[2]. However, reduced network activity can also impair specific aspects of health, notably wound healing[3,4]. This may be one reason why impaired wound healing is a common complication of diabetes[5]. Identification of the mechanisms mediating both the health benefits and impairments from reduced IIS and TOR activity is important in order to understand if they can be triaged apart.

The molecular and cellular mechanisms underlying impaired wound healing in diabetes are incompletely understood. Vascular defects have a major role[6], but insulin resistance in other cell types may also contribute and could be a potential therapeutic target. The fruit fly Drosophila is an excellent model organism in which to study the cell biology and physiology of epidermal wound healing. The fly has also been used to develop multiple genetic models of extended lifespan from reduced IIS/TOR activity, and of insulin resistance[1]. In several of these experimental situations, gene expression can be controlled both temporally and in a cell-type-specific manner.

The Drosophila epidermis is a monolayered epithelium with a basal lamina, resembling simple squamous epithelia in mammals. Altogether with the apically secreted cuticle it provides a robust external barrier (Supplementary Fig. 1A)[7]. The two major conserved mechanisms contribute to epithelial wound repair in the Drosophila embryo and pupa are formation of an actomyosin cable that constricts the wound, and protrusion of lamellipodia by cells surrounding the wound, associated with cell crawling towards it, eventually leading to wound closure[8–11]. In larvae, wound closure is driven primarily by actin protrusion rather than by actomyosin cable contraction, but a polarized actin assembly is observed at the wound edge[12–15]. The movement of cells surrounding the wound is coupled to phagocytosis of debris and formation of syncytia centred at the wound[12,16,17]. Haemocytes are also recruited to the wounded area, and these stop the escape of haemolymph and the entry of microorganisms[12,18].

The larval epidermis is post-mitotic. Epidermal morphogenesis occurs only during embryonic stages, and the larval epidermis hence provides a powerful in vivo model to study the cell biology of wound healing. Larvae have not been widely used in this context, partly because they cannot easily be immobilized for long periods. We have overcome this problem and established a laser ablation method for inducing epithelial wounds in third instar larvae. We employed live imaging to analyse the role of the IIS/TOR network, and found that inhibition of TORC1 impairs wound healing in an S6 kinase-dependent manner, and that there is an epidermal-specific and cell-autonomous requirement for insulin signalling through FOXO for efficient actomyosin cable formation and wound healing.

## Results

**Epidermal wound healing in Drosophila larvae.** A laser beam was used to ablate one or a few cells in the epidermis of anaes-thetized, early third instar (L3) Drosophila larvae, which were imaged for up to 7 h after wounding (see Methods). We initially investigated wounds encompassing 3–6 epidermal cells ($\sim$3,000–8,500 $\mu m^2$). Immediately after injury, the wound area expanded for 6 ± 1 min (Supplementary Fig. 1 and Supplementary Fig. 2A,B, Supplementary Table 1 and Supplementary Movie 1A). The sides of the cells facing the wound showed a strong enrich-ment of Src-GFP at the plasma membrane and formed a ring around the wound, which shrank over time. Accumulation of Src-GFP occurred near the apical side of the cell edges at the border of the wound while, at the basal side, lamellipodia emanated and filled in the wound. Wound closure was completed within 250 ± 20 min, irrespective of Gal4 drivers and markers used (Supplementary Fig. 1C–F and Supplementary Fig. 2A–D).

To investigate a possible role of a contractile actomyosin ring in the initial shrinking of the wound, we expressed DE-Cadherin-GFP to mark adherens junctions, and mCherry fused to spaghetti squash (Sqh), the Drosophila non-muscle myosin II regulatory light chain (MyoII), both under their endogenous promoters[19,20]. We saw that MyoII was rapidly reorganized in the cells surrounding the wound (Supplementary Fig. 1G and Supplementary Movie 1B,C), and relocated from the cytoplasm to the side of the cells facing the wound. There it co-localised with DE-Cadherin, which itself was then gradually lost from this location but remained unaltered at the cell sides facing non-wounded neighbours. Accumulation of MyoII occurred at the end of the expansion phase; a complete actomyosin cable had formed by 10–12 min and was maintained until wound healing was completed.

We next examined single-cell wounds, to explore the potential for investigating cell-autonomous effects. As in multicellular wounds, we observed an initial expansion of the wound, formation of an actomyosin cable and growth of basal lamellipodia into the wound (Fig. 1a–e, Supplementary Table 1 and Supplementary Movie 2). The rate of actomyosin cable formation was independent of wound size and number of ablated cells, but single-cell wounds healed faster (Supplementary Fig. 2E–H). Both in single- and multi-cell wounds, the actin cable was established by the cells surrounding the wound, with simultaneous depletion of cytoplasmic MyoII and LifeAct-Ruby in these cells (Fig. 1e, Supplementary Fig. 1G and Supplementary Fig. 3A–D). Extrusion of cells by a contractile actin cable also occurs after apoptosis. However, we did not see apoptotic markers[21] (Supplementary Fig. 3E), the ablated cells were immediately destroyed, and their debris was engulfed by neighbouring cells (see below). As in other systems[8,22], wounding was immediately followed by a pulse of Calcium signalling (Fig. 1f and Supplementary Movie 3).

Survival rates were higher after single-cell ablation (Fig. 1b, Supplementary Fig. 1D and Supplementary Fig. 4), and we therefore conducted our further investigations on single-cell wounds, which also allowed us to investigate cell autonomous effects of IIS/TOR signalling in the cells surrounding the wound.

**IIS during wound healing.** We first tested whether changes in IIS/TOR network activity occurred in cells around epidermal wounds. As a readout for PI3K activity, which is elevated on insulin signalling[23], we used the PIP3-sensor tGPH[23]. This marker was initially present throughout the cell, with an enrichment at the plasma membrane. On wounding, tGPH accumulated at the plasma membrane facing the wound area within 7 ± 1 min (Fig. 2a and Supplementary Movie 4), as previously reported in embryos[24]. The signal increased as lamellipodia grew into the wound area, decreased as the wound closed and then declined to undetectable levels 20–30 min after wound closure. IIS also results in translocation of FOXO from the nucleus to the cytoplasm. We used an mCherry knock-in in the endogenous dfoxo locus to visualize FOXO. Development and wound healing in foxo-mCherry larvae was normal. Before imaging and wounding, FOXO-mCherry was ubiquitously distributed throughout the cytoplasm and the nuclei of epidermal cells (Fig. 2b). Mounting and fluorescence microscopy led to an increase of the signal in the nucleus (Fig. 2b), suggesting that heat or laser scanning acts as a stress

stimulus. Within $8 \pm 1$ min after wounding, the cells directly around the wound began to lose FOXO-mCherry from their nuclei, and some cells in the second row also shuttled FOXO-mCherry from the nucleus to the cytoplasm (Fig. 2c-h and Supplementary Movie 5A–B). This distribution gradually reversed as FOXO re-entered the nuclei. IIS was thus activated by

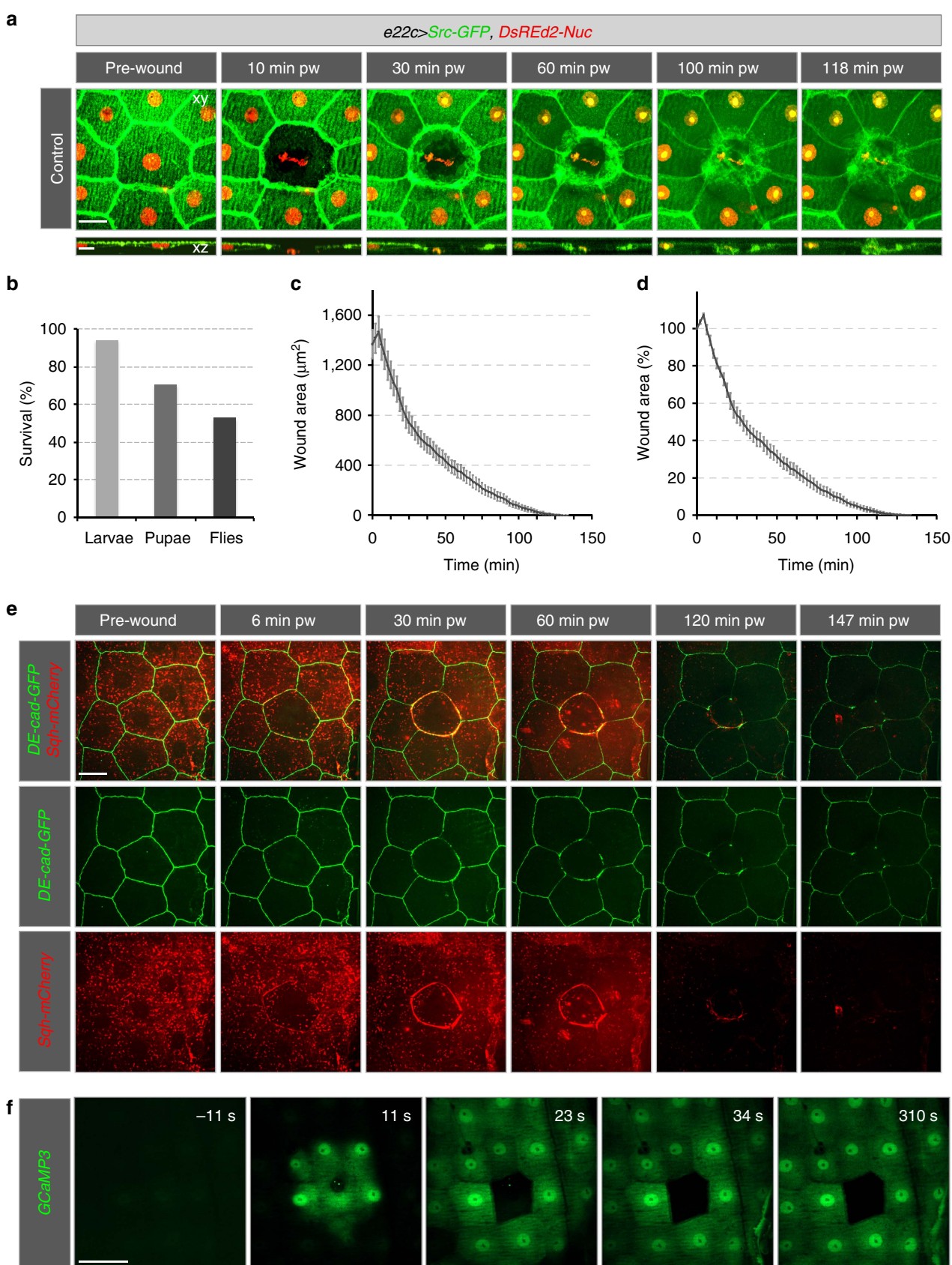

wounding in the cells surrounding the wound and the transcriptional activity of FOXO thereby repressed.

IIS/TOR signalling represses autophagy. We measured the responses of the commonly used autophagosome markers GFP-Atg8a and eGFP-huLC3 (Fig. 3a, Supplementary Fig. 5A–D and Supplementary Movie 6). Both markers showed a diffuse cytoplasmic distribution in control larvae, and both changed their distribution in the cells surrounding the wound. Unlike the other responses we studied, they showed changes only towards the end of the healing process, following the relocation of FOXO to the nucleus. Punctate accumulations, presumably representing autophagosomes, began to appear after ∼80 min, possibly associated with the digestion of ingested material. When we marked cells in different colours using the Brainbow system[25], we observed the debris of the dead cell being ingested by the epithelial cells that surrounded the wound (Fig. 3b and Supplementary Movie 7), as has also been observed in mice[26].

Autophagosomes appeared in cells up to four cell diameters away from the wound. Cells imaged simultaneously in a neighbouring segment (∼18–20 cells distant from the wound area) did not show this response (Supplementary Fig 5C–F), indicating that it was not caused by general stress resulting from the experimental manipulations.

In conclusion, events typical for IIS occurred in the cells adjacent to the wound during healing. We therefore tested whether the IIS/TOR network was required for normal wound healing.

**Ubiquitous reduction of IIS and TORC1 signalling**. We first reduced insulin signalling ubiquitously. Deletion of three insulin ligands (dilp2-3,5 −/−), as well as the expression of a dominant negative version of the insulin receptor (InR$^{DN}$) under the control of the ubiquitously expressed da-Gal4 driver delayed wound healing (Supplementary Fig. 6A–D). IIS is connected through multiple feed-forward and feed-back routes to TOR signalling. We used rapamycin to test whether TORC1 mediated wound healing and found that treatment with 1 or 20 μM rapamycin led to delayed wound healing (Supplementary Fig. 6E–G). Thus the normal wound healing response requires both IIS and TORC1 signalling.

**Effect of IIS and TORC1 on wound healing**. We next determined whether the observed delays in wound healing were attributable to a systemic requirement for IIS, or whether normal IIS was required within the epidermal cells themselves. Expressing InR$^{DN}$ specifically in the epidermis resulted in a similar delay in wound closure, as seen after ubiquitous down-regulation of insulin signalling both for single-cell and multi-cell wounds (Fig. 4a-c, and see below). Therefore, insulin signalling is required specifically in epidermal cells for efficient wound closure.

We also investigated the roles of TORC1 and TORC2 signalling in the epidermal cells using a dominant-negative version of TOR or RNA interference against Raptor and Rictor

with RNAi lines whose efficacy has been extensively validated in other systems[27,28]. Epidermal expression of a dominant negative version of TOR or RNA interference against Raptor led to larval or pupal lethality. Surviving larvae were smaller than controls and their epidermal cells showed abnormal cell shape and morphology, making it impossible to conduct wounding experiments under conditions comparable to controls or to rapamycin-treated larvae. This situation was not improved when the onset of expression of the transgenes was delayed by using Gal80$^{ts}$. In contrast, epidermal reduction of TORC2 signalling had no effect on wound healing (Supplementary Fig. 7A), suggesting that TORC2 plays no role, and the effect of TOR on wound healing is therefore indeed mediated by TORC1.

To understand how the insulin receptor and TORC1 mediate wound healing, we identified their downstream molecular effectors. To test whether the loss of nuclear FOXO in the cells adjacent to wounds was required for normal wound healing, we manipulated the levels of FOXO either globally or in epidermal cells. Loss of FOXO (foxo$^{Δ94}$, A58 > foxo$^{RNAi}$or e22c > foxo$^{RNAi}$) had no effect on either single-cell or multi-cell wound healing (Fig. 5a–c, Supplementary Fig. 8A–B, Supplementary Fig. 9A–B and Supplementary Fig. 10A–B). Larvae expressing elevated levels of FOXO in the epidermis showed no morphological defects and the over-expressed FOXO reacted to wounding in the same manner as the endogenous FOXO, although nuclear levels at late stages even exceeded those seen before wounding (Fig. 6a–c and Supplementary Movie 8A-B). However, wound closure was significantly delayed in these larvae (Fig. 5a–c and Supplementary Fig. 9A–B and Supplementary Fig. 10A–B) showing that elevated levels of FOXO interfered with wound healing.

To distinguish between the effects of raised levels of FOXO in the nucleus or in the cytoplasm we expressed a mutant version of FOXO that is insensitive to insulin signalling (FOXO-TM)[29], and is therefore constitutively present in the nucleus. Overexpression of FOXO-TM in the epidermis at 25 °C leads to larval lethality. However, at 18 °C some larvae survived and wounding experiments showed significant delays in healing, with 67% of the wounds not closing at all (Supplementary Fig. 11A–B). This indicates that FOXO in the nucleus delayed wound healing.

We investigated whether FOXO mediates the delayed wound healing observed in IIS mutant larvae by reducing FOXO levels in larvae expressing InR$^{DN}$ in the epidermis. Reduction of FOXO by half (foxo$^{Δ94/+}$), as well as complete loss of FOXO (foxo$^{Δ94/Δ94}$) and foxo epidermal knockdown (foxo$^{RNAi}$) all suppressed the defects in wound healing caused by InR$^{DN}$ (Fig. 5a–c, Supplementary Fig. 8A–B, Supplementary Fig. 9A–B, Supplementary Fig. 10A–B and Supplementary Movie 9). In conclusion, suppression of FOXO activity in the nucleus of the epidermal cells is required for proper wound closure, and this suppression is lost upon over-expression of FOXO or reduced IIS.

We showed that TORC1 is required for normal wound healing. TORC1 activates the S6 kinase (S6K) and inhibits the translational inhibitor 4E-BP. To test the roles of these molecules, we

**Figure 1 | Healing of single-cell epidermal wounds in 3rd instar larvae.** (**a–d**) Heterozygous e22c > Src-GFP,DsRed2-Nuc (e22c-Gal4, UAS-Src-GFP, UAS-DsRed2-Nuc/ + ), (**e**) recombinant DE-Cad-GFP, Sqh-mCherry and (**f**) e22c > GCaMP3 (e22c-Gal4, UAS-GCaMP3) early third instar larvae were immobilized and wounded by laser. (**a**) Projections (xy) and cross-sections (xz; stacks) of the wounded epidermis from a time-lapse series of a single-cell wound that was closed by 118 min. (**b**) Survival rate of larvae after wounding. After single-cell wounding 94% of the larvae survived 48 h, 71% pupated and 59% eclosed as adult flies. (**c–d**) Quantitative analysis of wound healing (n = 10 larvae). Data given as mean ± s.e.m. (**c**) Changes in absolute wound area (μm$^2$) and (**d**) in relative wound area (%). After single-cell ablation, the wound area first expanded for 6 ± 1 min. 50 ± 4% of the wound area was covered within ∼23 min and the remaining 50% closed within ∼95 min. (**e**) Actomyosin dynamics of single-cell wound healing of n = 6 larvae. At 6 ± 1 min an actomyosin cable started to form and was completed at 10–14 min and maintained throughout wound healing. (**f**) Response of the Calcium sensor GCaMP3 (green) to wounding. GCaMP3 is ubiquitously expressed in the epidermis. After wounding a calcium wave spreads from the wounded area. Scale bars, (**a,e**) 20 μm, (**a**) in xz section: 10 μm, (**f**) 30 μm. pw: post-wounding, DE-Cad: Drosophila E-Cadherin, Sqh: spaghetti squash, the Drosophila non-muscle myosin II regulatory light chain.

used S6K and 4E-BP loss- and gain-of-function mutants. Neither loss ($4E\text{-}BP^{null/null}$)[30] nor constitutive activity of 4E-BP (using 4E-BP$^{CA}$, a construct that extends life span)[31] affected wound healing, but reduction of S6K activity in the epidermis slowed the healing process (Supplementary Fig. 12A-D and Fig. 7b). Thus the effect of TORC1 activity on wound healing is epidermis-specific, and is at least partly mediated by S6K.

If TORC1 acts mainly through activating S6K, then a constitutively active form of S6K should reduce the deleterious effects of loss of TOR signalling on wound healing. We expressed a constitutively active S6K in the epidermis of rapamycin-treated larvae (Fig. 7a–c and Supplementary Movie 10), and this had no effect in control larvae, but completely suppressed the rapamycin-induced delay in wound healing. Thus, S6K activation

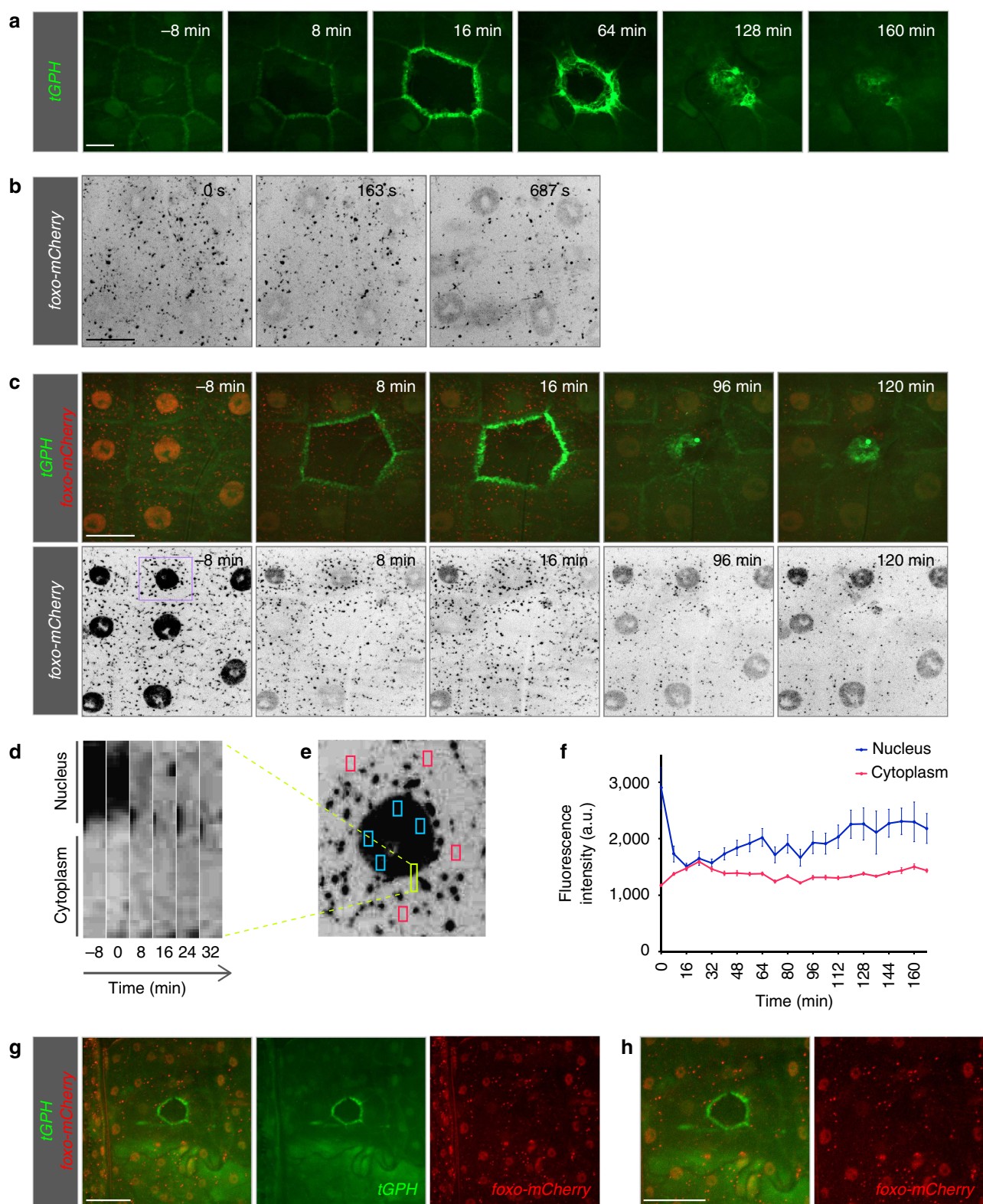

**Figure 2 | Insulin signalling during wound healing.** (**a**) Distribution of tGPH, a reporter for PIP3. tGPH, the PH-domain of GRP1 coupled to GFP, is expressed under the tubulin promoter in all tissues. At $7 \pm 1$ min, tGPH becomes polarized from an initially ubiquitous subcellular distribution in cells surrounding the wound. (**b–h**) Subcellular distribution of FOXO before wounding and during wound healing in L3 larvae in which mCherry was inserted into the *foxo* locus, generating a FOXO-mCherry gene product expressed under the endogenous *foxo* promoter. (**b**) Effect of mounting and imaging on FOXO distribution. When larvae were first mounted for imaging, FOXO-mCherry (grey) was ubiquitously distributed in the cytoplasm and in the nuclei of epidermal cells. Soon after mounting under the spinning disk confocal microscope and laser illumination (561 nm), FOXO-mCherry became enriched in the nuclei. The intense punctate pattern is seen with all mCherry constructs we have used in this study (see also Fig. 1e, for example). (**c**) Time-course of redistribution of epidermal FOXO-mCherry (red in upper panel and grey in lower panel) after wounding; the larvae also expressed tGPH (green). In cells directly surrounding the wound, nuclear FOXO began to translocate to the cytoplasm within $7 \pm 1$ min of wounding. By $16 \pm 1$ min nuclear FOXO had reached its minimum. (**d**) Kymograph of FOXO distribution in a region of a cell bordering the wound (yellow marked area in **e**). (**e**) Overview of the cell for which the analyses are shown in **d** and **f** with the analysed areas marked by coloured boxes. (**f**) Change in fluorescence intensity in four areas of the cytoplasm (red boxes in **d**) and nucleus (blue boxes in **d**) plotted against time. Mean ± s.e.m. (**g**) Lower magnification view of subcellular distribution of FOXO-mCherry (red) 10 min after wounding showing cells further beyond the wound edge. More distant cells did not respond to wounding and FOXO-mCherry remained nuclear. Scale bars, (**a**) 20 μm, (**b,c**) 30 μm and (**g,h**) 60 μm. L3: third instar larva.

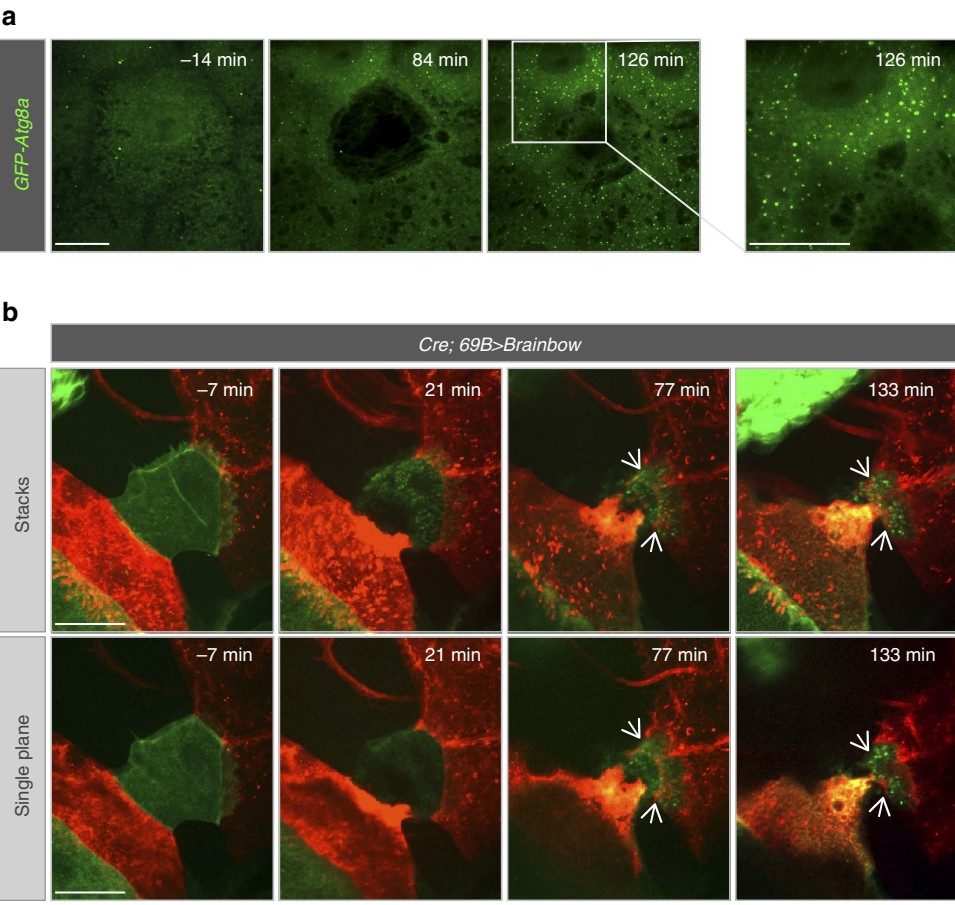

**Figure 3 | Autophagy during wound closure.** (**a**) Analysis of autophagy using *e22c > GFP-mCherry-Atg8a* (only the green channel is shown). Before wounding, the marker was seen in faint particulate structures throughout the cytoplasm. During wound healing the marker redistributed in cells surrounding the wound forming bright spots at later stages. (**b**) Epithelial cells phagocytose cell debris during wound healing. The Brainbow system was used to mark epidermal cells with different markers (GFP and RFP) by inducing *UAS-Brainbow-M* in epidermal cells (*Cre; 69B > Brainbow-M*). In the experiment shown here, a green cell was ablated cell. After wounding the green debris was engulfed (white arrows) by red epidermal cells surrounding the wound. The upper panels are a projection; the lower panels show a single plane to illustrate colocalisation of green and red fluorophores within the engulfing cell. Scale bars, (**a**) 20 μm, (**b** stacks and single plane) 30 μm. Transgene genotypes of larvae: *e22c > GFP-mCherry-Atg8a* (*e22c-Gal4, UAS-GFP-mCherry-Atg8a*) and *69B > Brainbow-M* (*Cre; 69B-Gal4,UAS-Brainbow-M*).

downstream of TORC1 is sufficient to enable efficient wound healing.

In contrast, S6K$^{CA}$ did not suppress the delay in wound healing caused by InR$^{DN}$ (Fig. 5a–c), suggesting that activation of FOXO is the main effector in this part of the signalling network.

In summary, the nutrient-sensing signalling network affects wound healing through two separate and parallel signalling branches, one acting through TORC1 and S6K, and the other through FOXO.

**Epidermal glycogen homoeostasis.** Glycogen in the larval fat body and muscles is tightly regulated by insulin/PI3K/AKT[32,33], and glycogen is also present in embryonic epithelial cells[34]. We

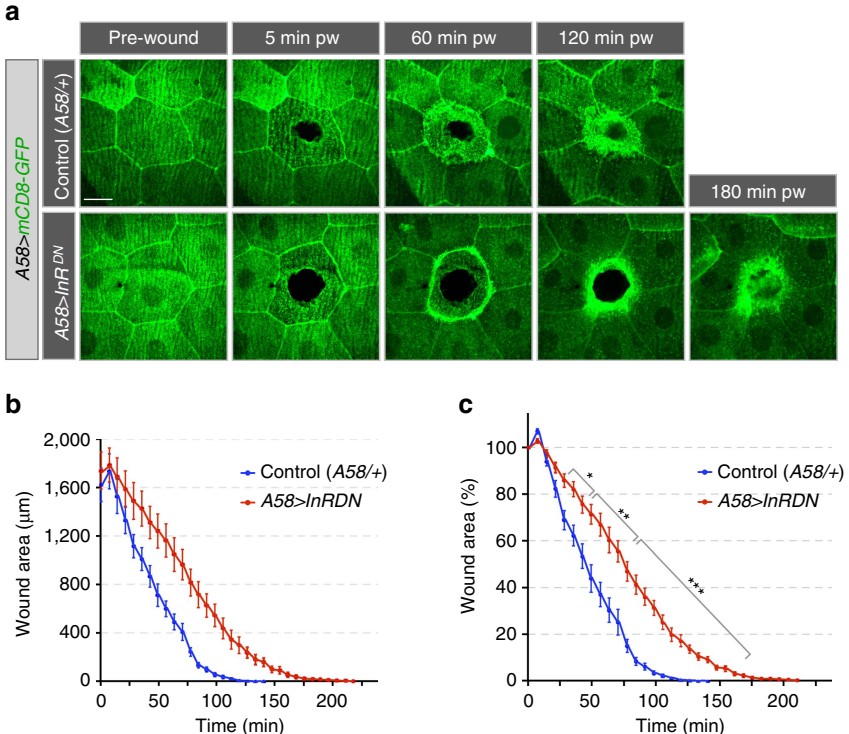

**Figure 4 | Effect of epidermal reduction of insulin signalling on wound healing.** (**a–c**) Wound healing in larvae expressing InR[DN] in the epidermis. (**a**) Time-lapse images of wound healing in L3 larvae. (**b–c**) Kinetics of wound closure. Time course of wound closure in (**b**) absolute ($\mu m^2$) and (**c**) relative (%) wound area. In controls ($n = 10$) wound closure was completed by $112 \pm 3$ min and in $A58 > InR^{DN}$ ($n = 9$) by $189 \pm 3$ min. (**b,c**) Mean $\pm$ s.e.m. two-tailed Student's $t$-test, $*P < 0.05$, $**P < 0.005$ and $***P < 0.0005$. Scale bars, 20 $\mu$m. Transgene genotypes of larvae: Control ($A58$-Gal4, UAS-mCD8-GFP/ $+$ ) and $A58 > InR^{DN}$ (UAS-InR[DN]; $A58$-Gal4, UAS-mCD8-GFP).

observed highly branched glycogen in epidermal cells of control third instar larvae, and glycogen was substantially reduced in the epidermis, but not the fat body or muscles, of larvae expressing InR[DN] in the epidermis (Supplementary Fig. 13A–F). Impaired glycogen metabolism could thus be one reason for the slow wound healing in larvae with lowered IIS. However, glycogen levels were normal in larvae expressing elevated levels of FOXO in the epidermis, and loss of FOXO did not restore the glycogen level in the epidermis of InR[DN] larvae (Supplementary Fig. 13E–G). Since elevated FOXO signalling interferes with wound healing, but not with glycogen storage, the impaired wound healing in FOXO-over-expressing larvae cannot be explained by defects in glycogen metabolism, at least under normal nutritional conditions. Raised levels of FOXO therefore probably affect wound healing independent of glycogen levels. Similarly, removal of FOXO ($foxo^{\Delta 94}$) restores both single and multi-cell wound healing but not glycogen levels in larvae expressing $InR^{DN}$ (Supplementary Fig. 13G, Supplementary Fig. 8A–B, Supplementary Fig. 9A–B, Supplementary Fig. 10A–B and Fig. 5a–c). Thus, wound healing can occur at a normal rate in the absence of normal levels of glycogen.

**Cell autonomy of insulin/FOXO signalling.** Insulin signalling regulates epidermal wound healing in a tissue-autonomous manner, and the signal could be needed in the cells that are damaged or in the surrounding cells that repair the wound. We created mosaics of cells expressing $InR^{DN}$ or $foxo$ in the epidermis.

When we ablated normal cells (that is, ones that did not over-express InR[DN] or FOXO) that were surrounded by three or more cells expressing InR[DN] or FOXO (marked by GFP expression), wound closure was delayed to a similar extent as in cases where all

surrounding epithelial cells expressed InR[DN] or FOXO (Fig. 8a–e). Thus insulin/FOXO signalling is required in the surrounding cells. If only one or two of the surrounding cells expressed InR[DN] or FOXO, then the rate of wound closure was normal (Fig. 8d–e). Thus, if the majority of the surrounding cells were normal, they could close the wound and compensate for the failure of a small number of cells that are unable to participate normally.

**Effect of IIS and FOXO on PIP3 and actomyosin assembly.** IS increases production of PIP3 in the plasma membrane, which then serves as a docking site for signalling molecules. Accumulation of the PIP3-reporter tGPH was both delayed and significantly weakened in larvae expressing InR[DN] (Fig. 9a,b) but normal in larvae expressing raised levels of FOXO (Fig. 9a,b).

Because PIP3 can recruit adaptors and activators of actin, we analysed the effect of IIS on the assembly and contraction of the actomyosin cable. We found that expression of InR[DN] and FOXO in the epidermis affected the actomyosin cable in the same manner. The cable formed later, was less pronounced and contracted more slowly (Fig. 9c–e and Supplementary Movie 11A). Thus, there was no correlation between PIP3 levels (reduced in $A58 > InR^{DN}$ but not in $A58 > foxo$) and assembly of the actomyosin cable (reduced in both cases), suggesting that the actomyosin cable does not strictly depend on high levels of PIP3 in the membrane. Therefore, IIS is unlikely to control the actomyosin cable through an effect of PIP3 on actin. Instead, InR[DN] and overexpression of FOXO both interfere with actomyosin cable formation, stability and rate of contraction by raising levels of FOXO in the nucleus.

In summary, reduction of insulin receptor signalling and over-expression of FOXO both inhibited epidermal wound healing, in

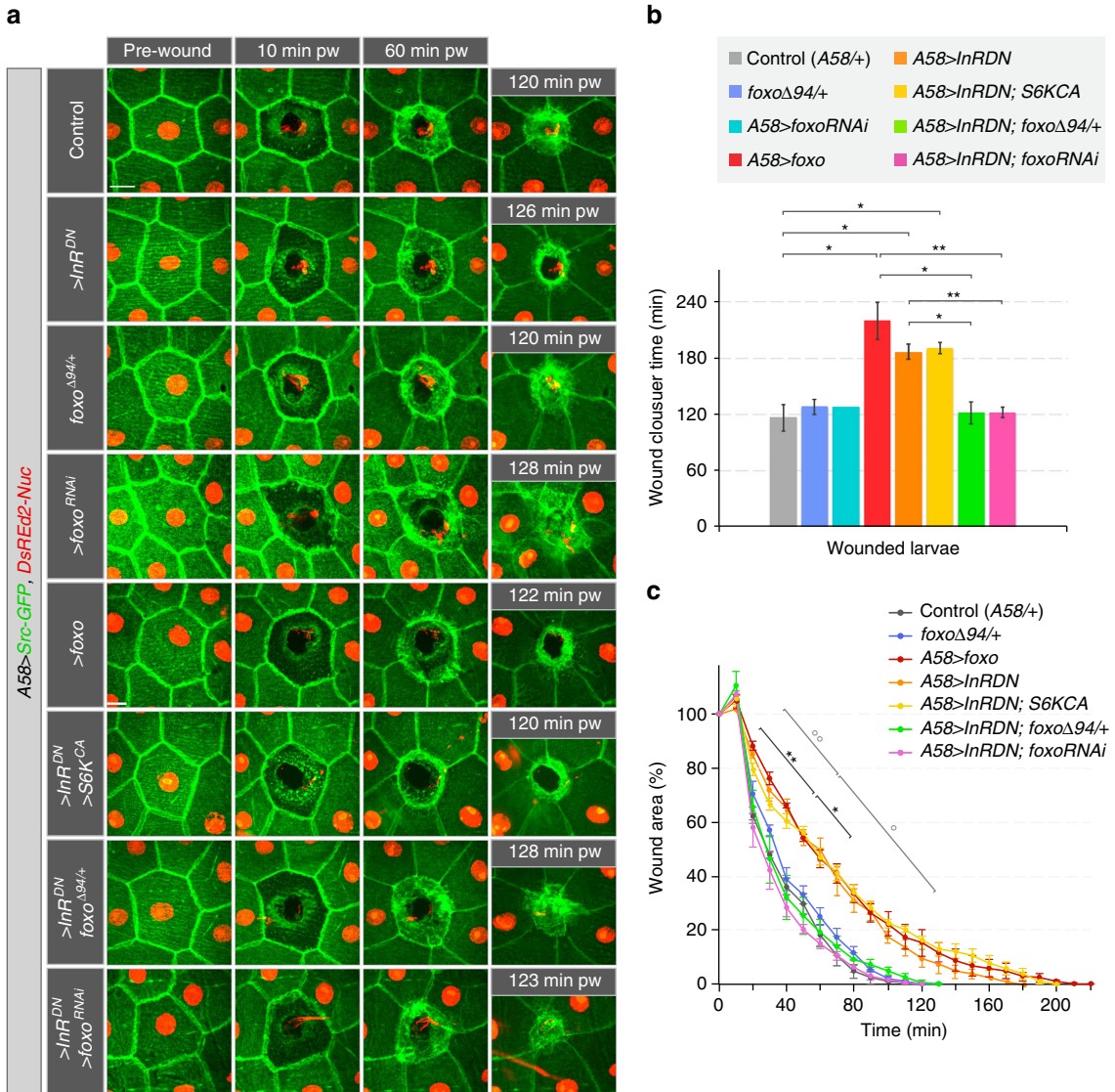

**Figure 5 | Loss of FOXO suppresses wound healing delays in larvae expressing InR^DN.** (a–c) Wound healing in larvae with modified InR, FOXO or S6K activity.(a) Time-lapse images of wound healing in L3 larvae. (b) Average time of wound closure for genotypes shown in a: 117 ± 20 min for control, 128 ± 11 min for $foxo^{\Delta94}/+$, 128 min for $>foxo^{RNAi}$, 220 ± 32 min for $>foxo$, 187 ± 12 min for $>InR^{DN}$, 188 ± 9 min for $>InR^{DN}$, $S6K^{CA}$, 122 ± 18 min for $>InR^{DN}$, $foxo^{\Delta94}/+$ and 122 ± 9 min $A58>InR^{DN}$, $foxo^{RNAi}$ larvae. (c) Kinetics of wound closure. Time course of changes in relative (%) wound area. (b,c) Mean ± s.e.m. one-way ANOVA with *post hoc* test *$P < 0.007$, **$P < 0.0007$ and ***$P < 0.00007$, $n = 3$–5 larvae each genotype. Scale bar: (a) 20 μm. Transgene genotypes of larvae: (a–c) All larvae carried $UAS$-$Src$-$GFP$ (green) and $UAS$-$DsRed2$-$Nuc$ (red). Control ($A58$-$Gal4/+$), $foxo^{\Delta94}/+$ ($A58$-$Gal4$, $foxo^{\Delta94}/+$), $>foxo^{RNAi}$($A58$-$Gal4$, $UAS$-$foxo^{RNAi}$), $>foxo$ ($A58$-$Gal4$, $UAS$-$foxo$), $>InR^{DN}$($A58$-$Gal4$, $UAS$-$InR^{DN}$), $>InR^{DN}$, $>S6K^{CA}$($A58$-$Gal4$, $UAS$-$InR^{DN}$, $UAS$-$S6K^{STDETE}$), $>InR^{DN}$, $foxo^{\Delta94}/+$ ($A58$-$Gal4$, $UAS$-$InR^{DN}$ $foxo^{\Delta94}/+$) and $>InR^{DN}$, $>foxo^{RNAi}$($A58$-$Gal4$, $UAS$-$InR^{DN}$, $UAS$-$foxo^{RNAi}$).

both cases because of elevated levels of FOXO in the nuclei of the epidermal cells. Failure of accumulation of PIP3 is a consequence of disrupted insulin signalling (either chronic or acute), but normal levels of PIP3 do not result in normal actomyosin cable formation if nuclear levels of FOXO are elevated. Lowered insulin signalling thus results in impaired wound healing because it leads to nuclear accumulation of FOXO, rather than affecting cable assembly through other routes.

Another important conclusion follows from these observations. In our experimental system the genes we over-expressed or knocked down were raised or lowered in expression throughout larval life. We were therefore unable to distinguish whether the effects on the duration of wound healing in the third larval instar were due to sustained disruption of IIS during the preceding larval instars, or whether they are due to a requirement of the

gene products at the time of wounding. However, some more specific conclusions about timing can be drawn. FOXO-shuttling occurred immediately after wounding, and presumably required insulin signalling at that time point. By contrast, the formation of the actomyosin cable was affected by the elevation of FOXO in the nucleus, the transcriptional consequences of which cannot have a direct effect within minutes. FOXO therefore must instead act through affecting the physiology of the cells during larval development prior to wounding. We cannot distinguish whether the effect of IIS on PIP3 accumulation is also due to prior effects, or to an acute need at the time of wounding. Thus sustained impairment of IIS activity in larval life presumably compromised the cells' ability to elaborate a proper actomyosin cable.

Surprisingly, loss of FOXO ($foxo^{\Delta94/\Delta94}$; $Sqh$-$mCherry$) also affected cable formation. While it did not affect the timing of

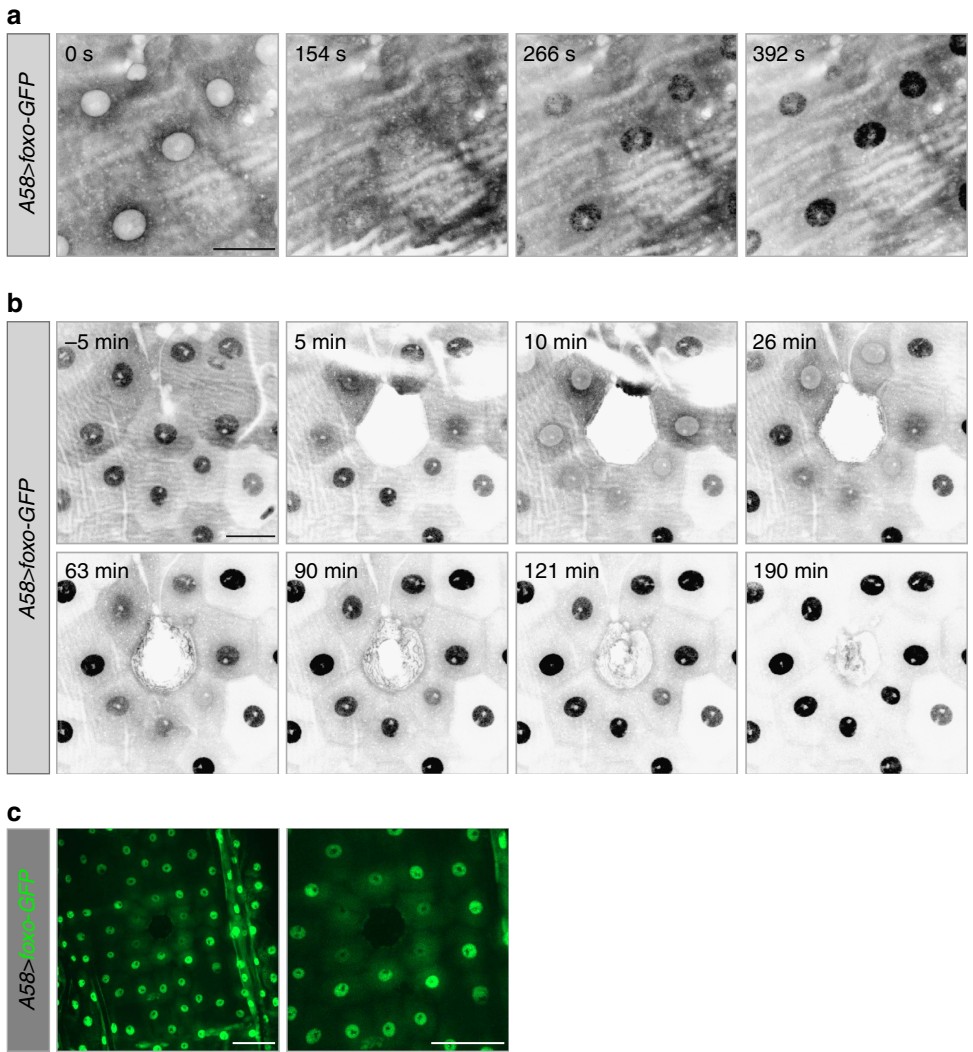

**Figure 6 | Subcellular distribution of overexpressed FOXO in the epidermal cells.** (a–c) FOXO distribution before and after wound healing in larvae overexpressing *foxo-GFP* in the epidermis (*A58>foxo-GFP*). (**a**) When larvae are first mounted for spinning disk confocal microscopy (488 nm laser), FOXO-GFP (grey) is present at higher levels in the cytoplasm than in the nucleus. FOXO-GFP then begins to be enriched in the nucleus, reaching maximal nuclear levels after ∼350–400 s; this also occurred when the larvae were observed only with epifluorescence, though at a slower rate. (**b**) On wounding most of the nuclear FOXO (grey) was shuttled into the cytoplasm (10–26 min). During the course of wound healing FOXO then again accumulated in nuclei, with high levels reached by ∼60–90 min. (**c**) Lower magnification view of subcellular distribution of FOXO-GFP (green) 12 min after wounding. Cells at a distance from the wound edge did not respond to wounding and FOXO-GFP remained nuclear. Scale bars, (**a**) 20 μm, (**b**) 30 μm and (**c** both panels) 60 μm.

cable assembly or the rate of contraction, the cable was significantly thicker (Fig. 9f–g and Supplementary Movie 11B), consistent with an inhibitory effect of FOXO on cable formation. We tested whether reduced insulin signalling affected wound healing through this effect of FOXO on the actomyosin cable. We found that in IIS deficient larvae lacking FOXO (*A58>InR^{DN}; foxo^{Δ94/+}*), the assembly and contraction of actomyosin cable was re-established to its normal rate (Fig. 9c–e), suggesting that IIS does affect wound healing through its effect on the actomyosin cable.

To address the mechanism by which FOXO affect actomyosin cable formation, we tested whether the compromised cable in larvae over-expressing InR^{DN} or FOXO could be repaired by supplying a constitutively active form of the myosin II regulatory light chain (Sqh^{E20E21}) (refs 35,36). In control larvae carrying *Sqh^{E20E21}*, the actomyosin cable was formed at the normal time and it contracted at the normal rate, but it was thicker, and failed to disassemble when the wound approached closure. Introducing *Sqh^{E20E21}* into larvae expressing InR^{DN} or raised levels of FOXO

in the epidermis (Fig. 10a–c) resulted in a cable that formed and contracted at the normal rate during the initial phase of closure, although again it was increased in thickness and blocked proper closure at late stages, confirming the notion that the primary effect of FOXO on wound healing is through the actomyosin cable.

These results show that a crucial role of the proper level of IIS in the larval epidermis is to inactivate FOXO to enable the cells to assemble a strong and functional actomyosin cable for efficient wound closure.

## Discussion

The capacity of the body to regenerate and repair tissues is compromised in some pathophysiological conditions, especially Type 2 diabetes[37]. Wound healing is also impaired by the TORC1 inhibitor rapamycin, which is licensed as a drug to prevent rejection of tissue transplants[38,39]. Consistent with these clinical findings, wound healing in mice depends on insulin and TOR

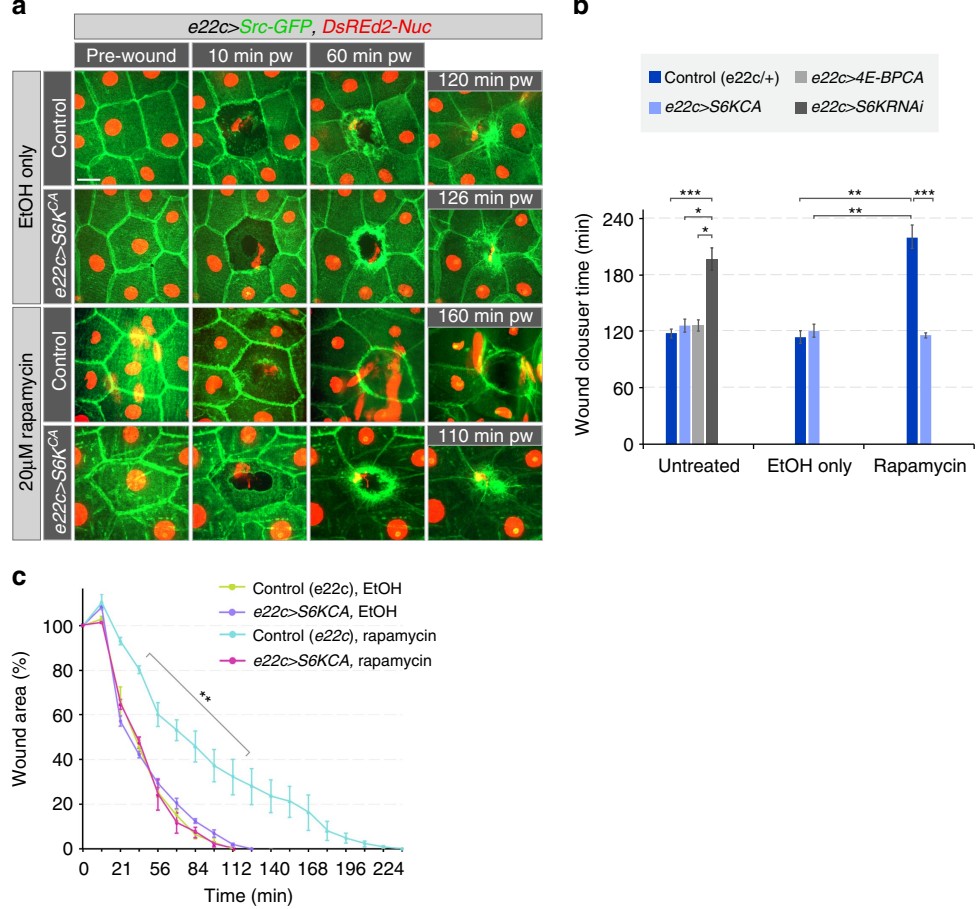

**Figure 7 | S6K activation suppresses rapamycin-induced wound healing delay.** (a–c) Wound healing in larvae expressing constitutively active S6K (S6KCA). All larvae expressed Src-GFP (green) and DsRed2-Nuc (red). (a) Time-lapse images of wound healing in L3 larvae. (b) Average time of wound closure for genotypes shown in a and Supplementary Fig. 12A: Untreated larvae: 118 ± 5 min for control, 126 ± 7 min for $e22c > S6K^{CA}$, 127 ± 5 min for $e22c > 4E-BP^{CA}$ and 196 ± 12 min for $e22c > S6K^{RNAi}$; larvae treated only with EtOH 114 ± 6 min for control and 120 ± 5 min for $e22c > S6K^{CA}$; larvae treated with 20 mM rapamycin in EtOH: 220 ± 12 min for control and 116 ± 2 min for $e22c > S6K^{CA}$. (c) Time course of changes in relative (%) wound area. (b,c) Mean ± s.e.m., (b) one-way ANOVA with *post hoc* test and (c) two-way ANOVA with *post hoc* test (comparing control and Rapamycin treatment). *$P < 0.01$, **$P < 0.001$ and ***$P < 0.0001$, $n = 3–5$ larvae each genotype and each treatment. Scale bar, (a) 20 µm. Transgene genotypes of larvae: Control ($e22c-Gal4/+$), $e22c > S6K^{CA}$ ($e22c-Gal4; UAS-S6K^{STDETE}$), $e22c > 4E-BP^{CA}$ ($e22c-Gal4; UAS-4E-BP^{CA}$) and $e22c > S6K^{RNAi}$ ($e22c-Gal4; UAS-S6K^{RNAi}$).

signalling[40–42]. However, the roles of systemic and cell autonomous signalling mechanisms are unclear, as are the exact processes of wound healing that are affected.

Drosophila is the organism that is best understood for mechanisms and pathways that direct tissue assembly, morphogenesis and cell polarity. Wounding larvae using laser ablation followed by long-term live imaging allowed us to characterize the activity of IIS with a high spatiotemporal resolution at the subcellular level. The large, transparent epidermal cells of Drosophila larvae, and the absence of interfering maternal components that may hamper genetic analysis of ubiquitously needed proteins in the embryo, make them an attractive model. Drosophila larvae can be treated with inhibitors and drugs (for example, rapamycin) with similar effects as observed in mammals[43]. Because many signalling pathways are conserved between Drosophila and mammals and many labs use Drosophila larvae to accelerate drug discovery for treatment in cancer or diabetes and other chronic diseases, this model may be useful for such purposes in the future[44–46]. Similarly, Drosophila larvae may be used to test the effects of dominant mutations of human genes to understand their mechanism and function, as recently described, for example, for the human keratin networks and their role in epidermolysis bullosa simplex[47].

Immediately after wounding we observed a local enhancement of IIS in the cells surrounding the wound and clonal analysis showed that IIS signalling during wound healing was autonomous to the epidermal cells themselves, and specifically to the small group of cells immediately around the wound. This is important, because it implies that very local insulin resistance, not only vascular complications peculiar to mammals, may contribute to impaired wound healing in type 2 diabetes, and that therefore local enhancement of IIS signalling at the wound itself could be beneficial.

A delay in epidermal wound healing after rapamycin treatment has also been observed in mammals[3,38]. In mice, the level of S6 phosphorylation (pS6) was increased in the epidermal compartment of wound edges, and rapamycin treatment reduced pS6 level and delayed the healing process[3,42]. Epidermal deletion of the TOR inhibitor TCS1 accelerated wound closure, with increased pS6 in the epidermis, in particular in the migrating epithelial tongue of the wound[3,42]. In Drosophila larvae, activation of S6K is sufficient to suppress the wound healing delay caused by rapamycin. Thus, the function of S6K in wound healing is conserved from insects to mammals. This conserved function makes S6K an interesting target for regenerative and drug studies in Drosophila to identify its cellular targets in wound healing.

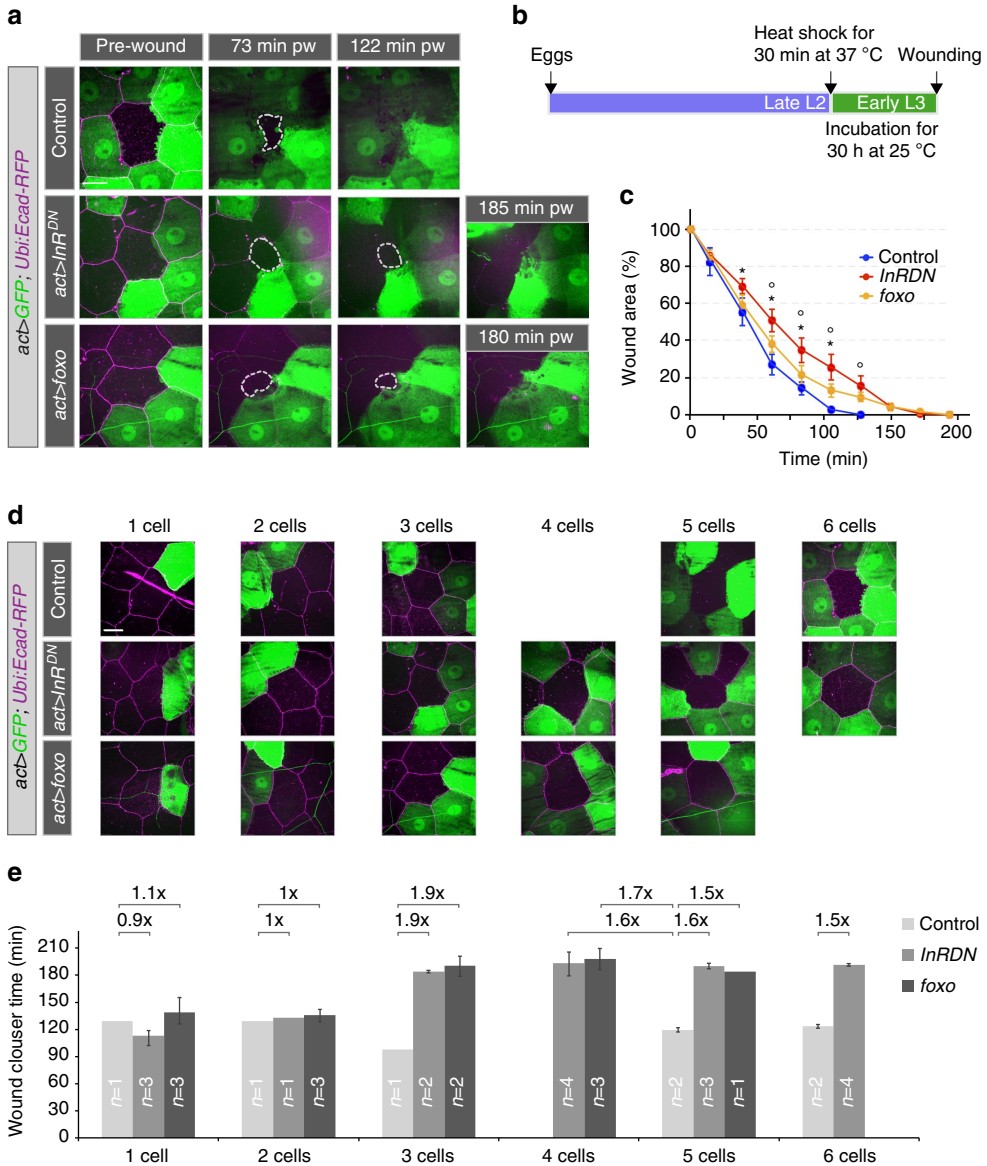

**Figure 8 | Clonal analysis of the requirement for insulin signalling.** (**a–e**) Wound healing in larvae with epidermal clones of modified InR and FOXO activity. Ubiquitously expressed *DE-Cad-RFP* was used to visualize all cells in the epidermis. (**a**) Time-lapse images of wound healing in L3 larvae. (**b**) Schematic for timing of transgene expression. (**c**) Kinetics of wound closure in larvae expressing InR^DN or FOXO in 3–6 cells surrounding a wounded cell. Time course of changes in relative wound area. Mean ± s.e.m. one-way ANOVA with *post hoc* test *$P < 0.01$ (control versus *InR^DN*) and °$P < 0.01$ (control versus *foxo*), $n = 3$–5 larvae for each genotype. (**d**) Examples of single-cell wounds with varying numbers of cells expressing InR^DN or FOXO or only GFP as their neighbours. (**e**) Average time of wound closure for mosaic genotypes shown in **d**. Mean ± s.e.m., 'n' is the number of larvae analysed in each case. Scale bars, (**a,d**) 20 μm. Transgene genotypes of larvae: Control (*hsflp; act > y + > Gal4, UAS-GFP/ + ; DE-Cad-RFP/ +*), *act > InR^DN* (*hsflp; act > y + > Gal4, UAS-GFP/UAS-InR^DN; DE-Cad-RFP/ +*), *act > foxo* (*hsflp; act > y + > Gal4, UAS-GFP/UAS-3xflag-foxo; DE-Cad-RFP/ +*).

Activation of S6K did not suppress the wound healing delay caused by reducing insulin receptor signalling, suggesting that other branches of insulin signalling in parallel to TOR are needed for efficient wound healing. This fits with the finding that reduction of FOXO levels, whether ubiquitously or only in the epidermis, completely rescues the wound healing delay caused by lowered insulin receptor signalling. Thus, at least two pathways within the nutrient-sensing network are involved in wound healing, one acting downstream of TORC1 through S6K and independent of FOXO levels, the other downstream of the insulin receptor through FOXO. The effects of modulating FOXO levels in mice are less clear. Loss of epidermal FOXO1 has been reported to lead either to impaired wound closure[48] or to improve wound healing[4,49]. Our own results clearly show the need for low nuclear FOXO levels for efficient wound healing in the Drosophila epidermis. Microarray experiments on wounded versus non-wounded human epidermis revealed an over-representation of FOXO binding sites in differentially expressed genes[50,51]. The genes repressed by FOXO included those encoding the matrix metalloproteases MMP-1 and MMP-9, which is interesting in light of the fact that Drosophila MMP-1 and MMP-2 are required for proper wound healing in larvae[52]. Cutaneous wound healing is difficult to study at the subcellular level in humans due to the interplay of several cell types and growth factors. The potential common FOXO targets between Drosophila and humans again highlight the usefulness of Drosophila as a model for further studies on the mechanism of IIS/FOXO regulation during wound healing.

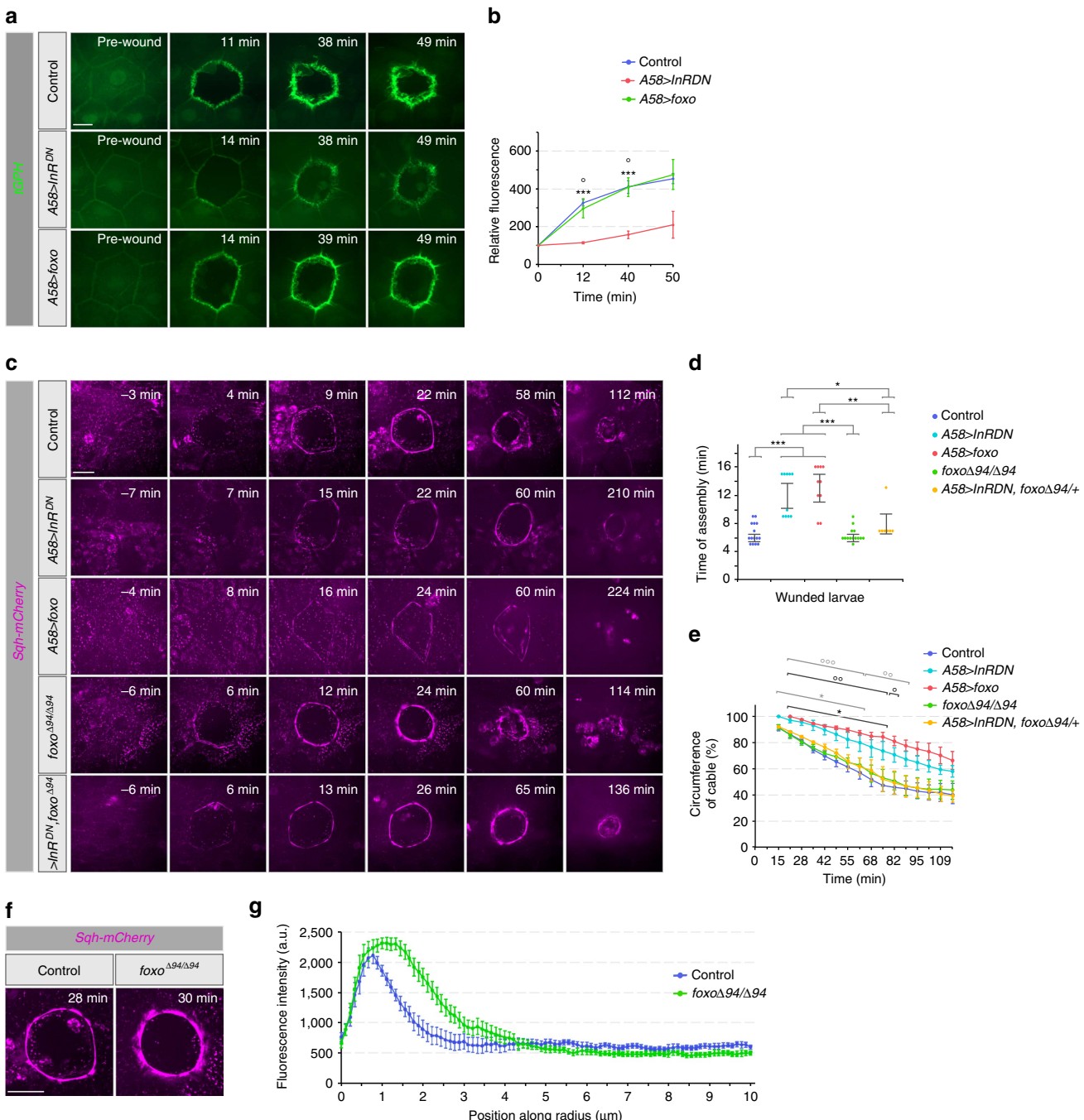

**Figure 9 | Effect of FOXO levels on PIP3 and actomyosin cables.** (**a–b**) PIP3 accumulation and (**c–g**) actomyosin cable formation during wound healing in larvae with modified levels of InR signalling and FOXO. (**a**) tGPH distribution during the early phase of wound healing. (**b**) Time course of fluorescence changes in wound edges relative to $t = 0$ (set at 100). Mean ± s.e.m. one-way ANOVA with *post hoc* test ***$P < 0.0001$ for control versus $A58 > InR^{DN}$ and °$P < 0.01$ control versus $A58 > foxo$, $n = 4$ larvae each genotype. (**c**) Time-lapse images of actomyosin cable assembly during wound healing in L3 larvae. (**d**) Average time for initiation of actomyosin cable formation for genotypes shown in **c**: 6 min for control, 12 min for $A58 > InR^{DN}$, 13 min for $A58 > foxo$-GFP, 6 min for $foxo^{\Delta94/\Delta94}$ and 8 min for $> InR^{DN}, foxo^{\Delta94}$. Mean ± s.e.m. one-way ANOVA with *post hoc* test, *$P < 0.01$, **$P < 0.001$ and ***$P < 0.0001$. (**e**) Kinetics of actomyosin cable contraction. Time course of changes in relative cable circumference. Mean ± s.e.m. one-way ANOVA with *post hoc* test, *$P < 0.01$ (black * marks for control versus $A58 > InR^{DN}$ and grey * marks for control versus $A58 > foxo$), °$P < 0.01$, °°$P < 0.001$ and °°°$P < 0.001$ (black ° marks for $foxo^{\Delta94/\Delta94}$ versus $A58 > InR^{DN}$ and grey ° marks for $foxo^{\Delta94/\Delta94}$ or $A58 > InR^{DN}, foxo^{\Delta94}/+$ versus $A58 > foxo$. $n = 4$–5 larvae each genotype. (**f**) Micrographs of actomyosin cables in control and $foxo^{\Delta94/\Delta94}$, 28 and 30 min after wounding. (**g**) Quantification of cable thickness in control and $foxo^{\Delta94/\Delta94}$ wounds. For each wound, 6 radial lines were drawn and the fluorescent intensity plotted along a length of 10 μm of the radius, starting at ∼1 μm inside the actin cable. ($n = 3$ wounds for each genotype). Mean ± s.e.m. two-tailed Student's *t*-test, *$P < 0.05$ for distances of 0.9–1.3 μm and 2.9–3.7 μm and ***$P < 0.0005$. for 1.4–2.8 μm. Scale bars, (**a,c,f**) 20 μm. Transgene genotypes of larvae: (**a–b**) Control ($tGPH/+$; $A58$-$Gal4/+$), $A58 > InR^{DN}$ ($tGPH/+$ $A58$-$Gal4$, $UAS$-$InR^{DN}$), $A58 > foxo$ ($tGPH/+$; $A58$-$Gal4$, $UAS$-$foxo$). (**c–g**) Controls ($Sqh$-$mCherry/+$, $A58Gal4/+$), $A58 > InR^{DN}$ ($Sqh$-$mCherry/+$, $UAS$-$InR^{DN}$; $A58$-$Gal4$), $A58 > foxo$ ($Sqh$-$mCherry/+$; $UAS$-$foxo$-$GFP$, $A58$-$Gal4$), $foxo^{\Delta94/\Delta94}$($Sqh$-$mCherry/+$; $foxo^{\Delta94}/foxo^{\Delta94}$), $> InR^{DN}, foxo^{\Delta94}$ ($Sqh$-$mCherry/+$, $UAS$-$InR^{DN}$; $A58$-$Gal4$, $foxo^{\Delta94}/+$).

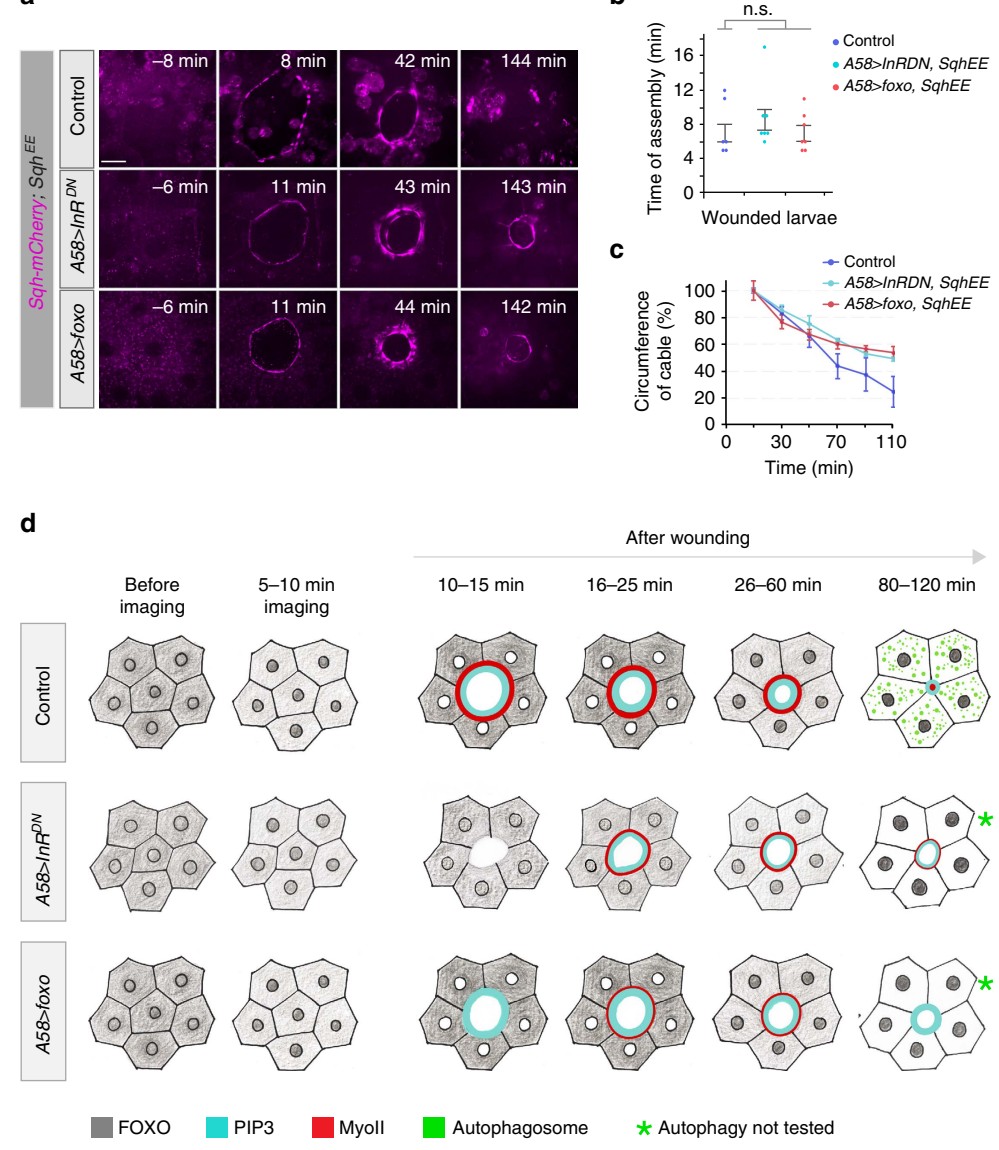

**Figure 10 | Effect of activated non-muscle myosin II regulatory light chain on the actomyosin cable.** (**a**–**c**) Actomyosin cable formation during wound healing in larvae with modified MyoII (Sqh), InR or FOXO activity. These experiments were done in parallel to the experiment in Fig. 9c-e. (**a**) Time-lapse images of actomyosin cable assembly during wound healing. (**b**) Average time for initiation of actomyosin cable formation for genotypes shown in **a**: 7 min for control, 8.5 min for $A58 > InR^{DN}$ and 7 min for $A58 > foxo$-GFP. (**c**) Kinetics of actomyosin cable contractions. Time course of changes in relative cable circumference. Mean ± s.e.m. one-way ANOVA with *post hoc* test, $n = 3$ larvae each genotype indicating non-significant difference in **b** and **c**. (**d**) Cartoon of events after wounding in normal and IIS-compromised epidermis, showing subcellular distribution of FOXO (grey), PIP3 (cyan), MyoII (red) and autophagsomes (green). Autophagy was not tested in $A58 > InR^{DN}$ and $A58 > foxo$-GFP (asterisk). Scale bar, (**a**) 20 μm. Transgene genotypes of larvae: (**a**–**c**) Controls ($Sqh$-mCherry/ + ; A58-Gal4, $Sqh^{E20E21}$/ + ) $A58 > InR^{DN}$($Sqh$-mCherry/ + , UAS-InR$^{DN}$; A58-Gal4, $Sqh^{E20E21}$/ + ), $A58 > foxo$ ($Sqh$-mCherry/ + , UAS-foxo; A58-Gal4, $Sqh^{E20E21}$/ + ).

High levels of glycogen are present in the epidermis of larvae, and in the mammalian epidermis[53,54]. The reduced glycogen level under lowered insulin receptor signalling was not normalized after FOXO deletion, while the rate of wound healing returned to normal. Thus, in Drosophila larvae under rich nutrient conditions, wound healing does not depend on high levels of glycogen, and the effects of reducing IIS can therefore not be explained by loss of glycogen. Glycogen increases during wound healing in the mammalian epidermis[55] but its function for wound healing and epidermal homoeostasis is not clear. It will be interesting to find out how glycogen synthesis is regulated in the epidermis and in response to wound healing and if there is a feedback loop or cross talk between glucose metabolism, glycogen synthesis and autophagy.

Our genetic manipulations affected IIS over an extended period of >24 h before we assayed their effects on wound healing. The effects could therefore be due to a requirement for IIS at the time of wounding, shortly afterwards, or in the period before wounding. We can distinguish distinct phases in wound healing, and these may be affected differently. The immediate increase in PIP3 at the wound edges and the appearance of an actomyosin cable coincided with the cessation of wound opening during the early response; this was then followed by constriction of the wound concomitant with the production of lamellipodia; finally, in the late phase of wound healing, autophagosomes appeared. IIS appeared to have a role immediately after wounding, as shown by the initial release of FOXO from epidermal nuclei. While IIS is needed for the accumulation of PIP3, we cannot distinguish

whether this is due to acute InR signalling at the time of wounding, or an earlier requirement, with the direct signal for PIP3 accumulation being provided by a different mechanism—for instance via $Ca^{++}$ signalling, activation of Src kinase, or other pathways. The appearance of autophagosomes in the late phase of wound closure may be a consequence of FOXO re-entering nuclei after the initial response to wounding, but it could also be stimulated by other mechanisms, such as inflammatory signalling or production of ROS.

However, overexpressing FOXO-TM also affects some of the immediate responses to wounding. FOXO-TM must act via transcriptional regulation, as it is constitutively confined to the nucleus. Since the time between wounding and actomyosin cable formation is too short to be regulated through a transcriptional mechanism, this means that increased levels of FOXO act at an earlier phase to compromise the ability of the epidermal cells to generate a strong and properly assembled actomyosin cable.

Previous studies on wound healing after mechanical injury of the larval epidermis did not describe the immediate formation of an actomyosin cable. Instead, a discontinuous assembly of actin was seen in the leading edges of cells surrounding the wound, and only several hours after injury[13–15]. On the basis of this delayed appearance of the cable, Brock et al. suggested that actin-based contraction is not a major contributor to larval wound closure. This is a clear contrast to wound healing in the embryo where actomyosin cables are formed[8,10], and now also to the data we report here. Unlike the visualization of the actomyosin cable by live imaging used here, the other studies analysed fixed material stained with phalloidin or other probes for actin. It is possible that the early actomyosin cable is sensitive to fixation.

Elevated nuclear FOXO reduced the rate of actomyosin assembly, contraction and maintenance and decelerated wound closure, suggesting that this may be due to transcriptional regulation of actin modulators. A transcriptome analysis in adult flies identified FOXO target genes that encode components of cell adhesion and cell polarity complexes and the Rho-GTPase signalling system (for example, p120, α-Cat, Baz/Par3, Shg/E-cadherin, Numb, Crb, Tub, Rho-GAP-1A, Rho-GAP-54D, Rho-GAP-93B, RhoL, Rac2)[56]. Some of these genes are involved in wound healing in Drosophila or mammals[9,24,57].

The finding that insulin and TOR signalling have independent roles in wound healing is important. First, the finding implies that the mechanisms of impaired wound healing seen with rapamycin treatment and insulin resistance in type 2 diabetes may be different. Second, it will be important to identify the effector molecular processes at work downstream of both. Finally, lowered insulin signalling in wounded tissues themselves, as opposed to impaired vascular supply, may have an important role in the impaired wound healing in type 2 diabetes, and may imply that local up-regulation of insulin signalling or inhibition of FOXO is sufficient to effect an improvement.

## Methods

**Fly stocks.** Fly stocks were maintained at 25 °C under a 12:12 h light/dark cycle at constant 65% humidity. All stocks were in a white-eyed genetic background. For ectopic expression, the Gal4/UAS system was used[58]. We used the e22c-GaL4 and A58-Gal4 driver lines to express UAS-constructs in the epidermis[12,59]. In all experiments e22c-Gal4, UAS-Src-GFP, UAS-DsRed2-Nuc/CyO or A58-Gal4, UAS-mcD8-GFP/TM6B or A58-Gal4, UAS-Src-GFP, UAS-DsRed2-Nuc/TM6B[12] or da-Gal4[60] females were crossed with males of the desired genotype. Both lines without balancers were used as control. The UAS-transgenes were: UAS-InR^DN (InR^K1409A: BL 8251; 8252 and 8253), UAS-foxo^RNAi (VDRC v 107786), UAS-S6K^RNAi(BL 41895), UAS-S6K^CA (S6K^STDETE: BL 6914), UAS-GFP-mCherry-Atg8a (BL 37749), UAS-mCD8-GFP (BL 5137), UAS-GCaMP3 (BL 32116), UAS-LifeAct-Ruby (BL 35545), UAS-Apoliner (BL 32122)[21], UAS-4E-BP^CA (UAS-4E-BP^TA)[31], UAS-Raptor^RNAi (BL 34814 and BL 31529), UAS-Rictor^RNAi (BL 31388 and BL 36699), UAS-foxo-TM[29], UAS-foxo-GFP[61]

UAS-3xflag-foxo (Chr. II) and UAS-foxo (Chr. III)[62]. 4E-BP null allele (4E-BP^null/null)[30], dilp2-3, 5 mutant[63], tGPH (BL 8164)[23], tGPH; Sb/TM3 (BL 8163)[23], SqhE20E21[35,36], Cre; 69B-Gal4 and UAS-Brainbow-M (gift from Stefan Luschnig)[25]. endo:Sqh-GFP (gift from Eric Wieschaus) and endo:DE-Cad-GFP, endo:Sqh-mCherry (gift from Thomas Lecuit)[19,20] were used to visualize actomyosin cable formation. In the second line, spaghetti-squash-mCherrry was recombined with DE-Cadherin-GFP and both constructs were expressed under endogenous promoters. In rescue experiments, the foxo null allele, foxo^494, was used[64].

To generate the UAS-3xflag-foxo fly line the dfoxo ORF was PCR amplified using primers SOL400 (CACCATGGACTACAAAGACCATGACGGTGATTATAAAGAT CATGACATCGATTACAAGGATGACGATGACAAGATGATGGACGGCTACGG GC) and SOL383 (CTAGTGCACCCAGGATG GTGGC) and directionally cloned into the pENTR/D-Topo vector (Invitrogen). The SOL400 primer encodes an N-terminal 3xFLAG-tag in frame with the dfoxo ORF. The 3xflag-foxo insert was cloned via Gateway technology into the pUAST gattB vector and the DNA was integrated into the genome by PhiC31-mediated transgenesis into the attP40 site[65]. The dfoxo-v3-mCherry knock-in line was generated by genomic recombineering[19] of the endogenous dfoxo locus in a two-step process (Supplementary Fig. 14). First, the 3′ end of the dfoxo gene encompassing exons 9–10, 12–14 and 16 of FBgn0038197 (Flybase release r6.07) was replaced by an attP-site using ends-out homologous recombination. The attP-site was then used to reinsert the foxo-mCherry C-terminal fusion construct using PhiC31-mediated transgenesis[19].

**Flip-out experiments.** InR^DN and foxo clones in the epidermis were generated with a flip-out system[66,67]. Female flies with the genotype hsFlp; Act5C > y + > Gal4, UAS-GFP/CyO were crossed with UAS-InR^DN; Ubi-DE-Cad-RFP/TM6B or UAS-foxo; Ubi-DE-Cad-RFP/TM6B males or in control experiments with Ubi-DE-Cad-RFP/TM3[68] males. Clones were identified by GFP expression under the Act5C promoter. Mating was carried out at 18 °C and at this condition no GFP was detected. The Flip-out was induced by heat shock (30 min at 37 °C) of mid-stage L2 larvae. The larvae were incubated for 30 h at 25 °C followed by wounding in third instar larvae (L3).

**Rapamycin treatment.** Rapamycin (LC Laboratories) was dissolved in 200 μl ethanol at a concentration of 1 or 20 μM and added to SYA food. For control food, 200 μl ethanol was added. 0.3% Erioglaucine disodium (food colouring) was used to monitor larval food uptake. Larvae were maintained till second instar larva (L2) on standard fly food (approximately 4 days AEL) and then switched to food supplemented with rapamycin for 21 h before the wound healing assay.

**Immobilizing larvae.** Early L3 larvae were rinsed with tap water (no longer than 5 min) dried carefully and anaesthetised using diethyl ether. Larvae were selected and transferred with a pain brush into a self-made eppi-cage[69] dried with a small piece of napkin. The eppi-cage was then inserted into the etherized chamber[69]. For long-term immobilization of larvae we anaesthetised them for 3.5 min with diethyl ether. 15–20 anaesthetised larvae were transferred into a cell culture dish with glass bottom (35 mm, Greiner bio-one art. no: 627861) and sorted and oriented under a stereomicroscope.

**Laser wounding.** Epidermal cells were wounded by laser on a spinning disk confocal microscope (Ultra-View VoX, PerkinElmer; Inverse, Nikon TiE) equipped with 355 nm pulsed ultraviolet laser (DPSL-355/14; Rapp OptoElectronic). Wounds were induced at the dorsal midline of abdominal segment A3 or A4 with 0.25–0.3 μJ pulsed energy in ∼1 ns. Laser ablation was conducted during time-lapse imaging.

We expressed fluorescent markers, RNAi and protein constructs using three Gal4 drivers. da-Gal4 is expressed ubiquitously[60], e22c-Gal4 in the epidermis from embryonic stage 10–11 onward[59], and A58-Gal4 in the epidermis from the first larval instar onward[12]. Only larvae with normal epidermal morphology and body size and that developed and survived to adulthood were used for evaluations and quantitative analyses. After wound closure, larvae were returned individually into food vials and monitored for survival. We evaluated data only from larvae that survived at least two days post injury

**Live imaging.** Larvae were imaged at ∼25 °C on a spinning disk confocal microscope (Ultra-View VoX, PerkinElmer; Inverse, Nikon TiE) with a ×60 Plan C-Apochromat water-immersion objective and an attached CCD camera (C9100-50 CamLink; 1000 × 1000 pixel) controlled by the Volocity software 6.3. 57 individual z-stacks with a step size of 0.28 μm were taken every 2–15 min for a 3–7 h.

**Image processing and quantification.** Images were processed using Fiji (National Institutes of Health) or Volocity 6.3. Wound areas, defined as the area left open between the lamellipodia, were measured manually with Fiji software during the early expansion phase, we measured the area left open by the dead cell.

**Transmission electron microscopy.** For transmission electron microscopy, Drosophila larvae were fixed and treated (cryo-immobilized by high-pressure freezing) as previously described[34,70].

**Statistics.** All data are presented as mean ± s.e.m. To estimate statistical significance one-way or two-way ANOVA plus *post hoc* test or two-tailed distribution Student's *t*-test was used. Statistical tests and *P* values are indicated in the figure legends. Statistical analyses were performed using Exel 2011.

**Data availability.** The authors declare that the data supporting the findings of this study are available within the article and its supplementary information files or are available from the corresponding author upon request.

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

## Acknowledgements

We are grateful to members of the Partridge and Leptin laboratories, especially M. Rembold, C. Lesch, U. Temp, M. Kakanj, J. Roth and L. Vogelsang for helpful discussions, comments and experimental assistance. We thank V. Riechmann, S. Roth and C.M. Niessen for critical reading of manuscript, M. Kakanj for graphical support, A. Schauss, C. Jüngst and N. Kladt from the CECAD imaging facility in Cologne and M. Flötenmeyer from the transmission electron microscopy imaging-facility of the MPI for Developmental Biology, Tübingen and the Bloomington and VDRC Stock Centres for fly strains. This work was supported by grants from the European Regional Development Fund and the German state North Rhine-Westphalia (NRW im Ziel 2) to S.A.E and L.P., a CMMC grant to M.L. and S.A.E. and a Boehringer Ingelheim Fonds fellowship to V.B.

## Author contributions

P.K., S.A.E., L.P. and M.L. conceived the project. P.K. designed and performed the experiments, developed the long-term live imaging and laser ablation for Drosophila larvae and analysed the data, except for the transmission electron microscopy, which was performed by P.K. and B.M. S.G. and V.B. generated the *foxo-v3-mCherry* and *UAS-3xflag-foxo* line. P.K., M.L., S.A.E. and L.P. drafted the manuscript.

## Additional information

**Competing financial interests:** The authors declare no competing financial interests.

