## [Peer review file · Nature Communications]

Reviewers' Comments:

Reviewer #1 (Remarks to the Author)

This a thorough study of how insulin and TOR signalling might regulate aspects of epidermal wound healing using the *Drosophila* larva as a model. When first describing their model they report how an actomyosin cable and lamellipodia appear to be closing the hole but don't cite the earlier papers describing this phenomena in the fly embryo (Wood et al NCB 2003 and Abreu-Blanco et al. JCB 2011)). More novel are their observations of FOXO shuttling between nucleus and cytoplasm, although the initial movement to the nucleus appears to be a slightly unnatural consequence of the imaging conditions; however, within 8 mins of laser wounding FOXO-mCherry relocates - in just the front two rows of cells - from nucleus to cytoplasm, and then 70 mins later the tagged FOXO returns to being ubiquitously localised within both cytoplasm and nucleus. At this time also the authors report accumulation of autophagosomes in cells up to four rows back from the wound edge. Both these phenomena are potentially downstream of IIS activity. And indeed it seems that TOR activity (specifically mediated by TORC2) is required for efficient wound closure in this model.

To test whether FOXO shuttling out of the nucleus is important for repair the authors overexpressed FOXO-TM (which causes FOXO to be insensitive to insulin signalling and to remain in the nucleus) in the larvae, and healing was retarded, but there are flaws with this experiment. First, remember that the nuclear FOXO is only there because the larvae is stressed by mounting and imaging, and second these larvae are the minority that escaped toxicity of the FOXO-TM even at 18°C. So how might the insulin signalling pathway be impacting on repair? The authors rule out glycogen storage because elevated FOXO disrupts healing without significantly altering glycogen levels in wound cells, although the role of glycogen and energy usage might be very different for bigger wounds. Indeed, the size of the wounds is one weakness of the model - the assay in this paper was simplified from a 3-4 cell diameter wound, down to a single cell ablation and one might argue that such a wound heals so rapidly that it barely models a skin wound on a fly, let alone a slow healing chronic wound on a diabetic patient. Indeed, it might be that these single-cell wounds behave more like single apoptotic epithelial cells, which are known to be extruded from the epithelium by an actomyosin-cable that forms in the surrounding epithelial cells (Rosenblatt et al 2001, *Current Biology*), rather than a multicellular wound.

The mosaic studies where DN^{InR} is expressed in a few cells at the wound margin are elegant, and talk to issues of cell autonomy, but to fully distinguish between a direct requirement of insulin and Tor signalling as part of the wound healing process versus a more general role in tissue homeostasis (which is clearly very relevant if one is to draw parallels with wound healing in patients with diabetes), the authors could try to block these pathways transiently e.g. by using *Da-Gal4+tub-Gal80ts +UAS-InR-DN* (or *UAS-Tor-DN*).

A general concern with this manuscript is that the numbers of wounds measured in most experiments is surprisingly low (1-5 wounds per genotype) - are these data statistically significant?

Reviewer #2 (Remarks to the Author)

This elegant study sets up the *Drosophila* larval epithelium as a model system to study the molecular requirements for efficient wound healing. This allows powerful *Drosophila* genetics to be applied to a medically relevant question, addressing issues relating to signaling, cell biology, cytoskeletal rearrangements, etc.

This study looks at the role of the insulin and TOR signaling pathways in wound closure. The finding that these signaling pathways are required for wound healing is not unexpected.

Surprising, instead, is the reason why. A priori, one would expect that insulin and TOR signaling are required acutely after wounding to promote cell growth. Instead, this study finds that the defective wound healing due to reduced insulin signaling can be rescued by removing FOXO. Since FOXO is a transcription factor, its effects are unlikely to take place within minutes - the timeframe of wound healing. Hence, it is likely that levels of insulin and FOXO signaling prior to the wounding predispose the cells for efficient or inefficient healing by modulating their transcriptional program. This is interesting and unexpected.

The data in this study are solid (and visually beautiful). The conclusions are well founded. This study is likely to be the first in a series of studies exploiting this system to study wound healing.

I have one main concern, which is that some of the results and implications are not presented or discussed clearly enough, leading to confusion or perhaps incorrect conclusions by the reader:

1.) Figure 2 shows nicely that within minutes after wounding, PIP3 levels increase and FOXO becomes cytoplasmic. However, since loss of FOXO rescues the defective wound healing caused by reduced insulin signaling (Figure 8), and since the transcriptional effects of FOXO are unlikely to translate into changes in cell behavior within 30 minutes, the authors conclude that the effects of FOXO are probably due to the chronic reduction of FOXO activity prior to the wounding experiment, and not to the acute inactivation of FOXO upon wounding. This also means, however, that it is unclear whether the acute increase in PIP3 levels shown in Figure 2 have any physiological relevance in this healing context. This should be made clear and explicit in the presentation of the study.

2.) Usually, increased autophagy correlates with (or is caused by) REDUCED IIS and TOR signaling, not INCREASED signaling. Hence, the elevated autophagy seen in Fig 2f clashes, or at least does not easily fit, with the elevated IIS signaling shown in Fig 2a. Even though the elevated autophagy is seen at 126 min after wounding, when PIP3 levels are returning to baseline and FOXO is becoming again more nuclear, there is nonetheless no indication (or data that I could see) suggesting that IIS or TOR signaling at this point have dropped BELOW the starting baseline levels, which could explain the autophagy induction. In the current phrasing, the authors try to gloss over this issue. This should be made more clear/explicit. It is very possible that the autophagy induction in this context is mediated by an inflammatory response or ROS, stimulating clean-up of debris, rather than anything having to do with IIS signaling, so unless there is evidence for reduced IIS signaling at this phase of the healing process, presentation of the autophagy data should be separated from the IIS data in the manuscript, both logically and structurally.

Minor Issues:

1. It is unlikely that localized production and/or release of dILPs at the site of wounding is causing the transient increase in PIP3 levels shown in Fig 2a. (It would have to be very fast, and act very locally. It's possible, but maybe not). Alternatively, I believe Src can activate PI3K cell autonomously? Could it be that activation of Src in the cells adjacent to the wound is leading to the increased PIP3 levels? If so, perhaps this possibility should be presented in the text. Otherwise, by presenting the dILP loss-of-function data in the following Figure 3, and by calling the signaling pathway "insulin/IGF signaling" gives the impression to the reader that the dILPs are likely playing a role in this context. Alternatively, is there reason to believe this effect is indeed mediated by dILP secretion?

2. Fig 2b: It would be helpful for the reader to see an image at lower magnification at 8m post wounding to show more clearly that indeed FOXO shuttles from nucleus to cytoplasm in the cells facing the wound, but not in cells further away, as stated in the text.

3. Fig 8a - is reduced accumulation of the PIP3 sensor upon InR[DN] expression due to reduced expression of the reporter? Or can this be excluded (e.g. with a GFP western from larval body walls?)

4. Fig 9a-c would need just A58>InR[DN] or A58>foxo as control genotypes to show rescue.

Reviewer #1

This a thorough study of how insulin and TOR signalling might regulate aspects of epidermal wound healing using the Drosophila larva as a model. When first describing their model they report how an actomyosin cable and lamellipodia appear to be closing the hole but don't cite the earlier papers describing this phenomena in the fly embryo (Wood et al NCB 2003 and Abreu-Blanco et al. JCB 2011).

Response:

We apologize for this omission. These obvious references had been included in our original version of the manuscript, but were then accidentally edited out during the final shortening of the text. We have now added them back.

More novel are their observations of FOXO shuttling between nucleus and cytoplasm, although the initial movement to the nucleus appears to be a slightly unnatural consequence of the imaging conditions; however, within 8 mins of laser wounding FOXO-mCherry relocates - in just the front two rows of cells - from nucleus to cytoplasm, and then 70 mins later the tagged FOXO returns to being ubiquitously localised within both cytoplasm and nucleus. At this time also the authors report accumulation of autophagosomes in cells up to four rows back from the wound edge. Both these phenomena are potentially downstream of IIS activity. And indeed it seems that TOR activity (specifically mediated by TORC2) is required for efficient wound closure in this model.

To test whether FOXO shuttling out of the nucleus is important for repair the authors overexpressed FOXO-TM (which causes FOXO to be insensitive to insulin signalling and to remain in the nucleus) in the larvae, and healing was retarded, but there are flaws with this experiment. First, remember that the nuclear FOXO is only there because the larvae is stressed by mounting and imaging, and second these larvae are the minority that escaped toxicity of the FOXO-TM even at 18oC. So how might the insulin signalling pathway be impacting on repair?

“nuclear FOXO is only there because the larvae is stressed by mounting”

Response:

This is largely correct. However, it does not invalidate the conclusion that upon wounding there is a signal that leads to shuttling of FOXO back out of the nucleus. FOXO is not entirely absent from the nucleus before imaging either – so there is a balance between nuclear and cytoplasmic FOXO, which is shifted to the nucleus upon imaging, and then back to the cytoplasm immediately after wounding, and finally back to the nucleus during late stages of wound closure.

We appreciate that the over-expression of FOXO or FOXO-TM is an artificial situation, but it nevertheless illustrates some of the consequences of manipulation of IIS. Tellingly, the endogenous tagged FOXO shows the same behaviour.

...these larvae are the minority that escaped toxicity of the FOXO-TM even at 18oC. So how might the insulin signalling pathway be impacting on repair?

Response:

High levels of nuclear FOXO are indeed toxic, and we therefore rely for our analyses on situations where the levels are just below the critical threshold. Since the experiments with FOXO-TM simply confirm the results obtained with the (less toxic) wild type FOXO, we do not see problems with the conclusions of this experiment, namely that it is the nuclear FOXO rather than elevated cytoplasmic FOXO that is responsible for the observed effects. More importantly, the biological significance of the role of FOXO is confirmed by a loss of function experiment (loss of FOXO suppresses the effects of loss of IIS). These results together support the role of FOXO activity in the nuclear compartment.

The authors rule out glycogen storage because elevated FOXO disrupts healing without significantly altering glycogen levels in wound cells, although the role of glycogen and energy usage might be very different for bigger wounds.

Response:

This is an interesting point, although we think it does not pertain here. Much larger wounds, which would take longer to heal, could also need more nutrients from the environment.

However, the value of the small wounds is precisely that they allow us to uncouple the requirements for IIS and for glycogen and provide us with a context where we can show that IIS has a function independent of glycogen. If very much larger wounds need glycogen, this would then be a requirement on top of the need for IIS seen for small wounds, and would not invalidate the glycogen-independent requirement for IIS for the wound healing process. However, we have now included experiments with larger wounds (80-120 μm diameter or 3000-8500 μm^2 area), and show that FOXO deletion completely suppresses the effect of InR^{DN} (new Supplementary Fig. 10). Since we know that glycogen is reduced or absent in epidermal cells expressing InR^{DN}, this result demonstrates that also larger wounds do not in fact rely on glycogen for proper healing, at least under nutrient-rich conditions.

Our multi-cell wound areas (80-120 μm diameter or 3000-8500 μm^2 area) and even some of the single cell wounds (40-60 μm diameter or 800-2500 μm^2) are larger than the multi-cell wounds studied by other authors using the embryonic epithelium (for example, Wood et al NCB 2003, Abreu-Blanco et al. JCS 2013, 20-40 μm diameter or 500-2500 μm^2) or multi-cell wounds in the larval (e.g. Galko and Krasnow 2004 PloS Biology; 100 μm diameter) and pupal epithelium (Anutunes et al., 2013 and Jacinto, Galko; 30-50 μm diameter), and the same size as multi-cell wounds induced in the adult fly (e.g. Losick et al. 2013; 2000-8000 μm^2).

Indeed, the size of the wounds is one weakness of the model - the assay in this paper was simplified from a 3-4 cell diameter wound, down to a single cell ablation and one might argue that such a wound heals so rapidly that it barely models a skin wound on a fly, let alone a slow healing chronic wound on a diabetic patient.

Response:

We agree that single cell wounds do not mirror long-term, non-healing wounds in diabetic patients. We present a reductionist approach in which a simplified system allows us to pick apart the various mechanisms that contribute to wound healing. Furthermore, some parasitic wasps lay their eggs in *Drosophila* larvae with wounds

that are about the size of those we induce here (Robertson et al PLoSOne 2013; the ovipositor that creates the wound has a diameter of 30µm diameter). Thus our wounds are of a physiologically relevant size.

Indeed, it might be that these single-cell wounds behave more like single apoptotic epithelial cells, which are known to be extruded from the epithelium by an actomyosin-cable that forms in the surrounding epithelial cells (Rosenblatt et al 2001, Current Biology), rather than a multicellular wound.

Response:

An interesting point, but we consider that this is not the case. We now include new data that show that cable formation occurs in the same manner in large and small wounds (New Supplementary Fig. 3). The destroyed cell is not extruded and instead we observe in Brainbow experiments that the debris of the ablated cell is engulfed by the cells surrounding the wound as they close it and the actin cable constricts (New Fig. 3b).

The mosaic studies where DNinR is expressed in a few cells at the wound margin are elegant, and talk to issues of cell autonomy, but to fully distinguish between a direct requirement of insulin and Tor signalling as part of the wound healing process versus a more general role in tissue homeostasis (which is clearly very relevant if one is to draw parallels with wound healing in patients with diabetes), the authors could try to block these pathways transiently e.g. by using Da-Gal4+tub-Gal80ts +UAS-InR-DN (or UAS-Tor-DN).

Response:

This is an important point. However, it is, unfortunately, impossible to do the ideal experiment with the tools that are currently available. We had already attempted to inactivate gene activity as close to the time of wounding as possible. But when we inactivated Gal80 within a period prior to wounding that would allow us to draw solid conclusions (e.g. six hours), we did not obtain sufficient levels of marker GFP expression. We have now repeated these experiments to test how long before wounding we must inactivate Gal80 in order to obtain sufficient signal. We find that with anything less than 24 hours we cannot do proper imaging. The expression levels allowed us to see cells and set wounds, but not to image for the periods of time needed for reliable quantification, because bleaching reduced the signal below detection level. The experiments were suggestive (i.e. visual inspection suggested the same effects as observed with longer-term expression of the transgenes), but they could not be quantified. Additional complications come from the fact that nutrient sensing signalling pathways are sensitive to temperature, so having to grow the Gal80^{ts}-carrying larvae at initially lower and then higher temperatures might also affect the results.

What this experiment would need would be a temperature sensitive protein that can be inactivated immediately by raised temperature (like *shibire*), or optogenetic tools for this signalling pathway, which we do not yet have.

Thus, we decided to be conservative in our interpretations of the results of these experiments. However, our clonal analysis also speaks to this point, so we know at least that the effects we observe depend on intact insulin signalling in the day before injury.

A general concern with this manuscript is that the numbers of wounds measured in most experiments is surprisingly low (1-5 wounds per genotype) - are these data statistically significant?

Response:

The data we present in the paper are those from movies in which we were able to image throughout the entire period of wound healing, without the larva moving or twitching, or drifting out of focus, or other disruptions, and where the imaging was of a quality that allowed proper quantification throughout this period. We have many other datasets that yielded qualitatively consistent results, but where at least one of the parameters of the experiment was not perfect. We have now included additional datasets from early experiments in which we used a different set of markers comparing small and large wounds, and we have also included these (New Supplementary Figs. 2, 9 and 10).

The one case where we present very small numbers, and do not have many other examples, are from the clonal analyses, and here the small sample sizes result from the fact that we have subdivided the cases into a range of very specific subgroups (New Fig. 8e). Since clones that can be evaluated are rare in the first place, special cases are obviously even more rare. However, all of the cases point in the same direction, and if all samples with 3 and more cells are evaluated together (as in Fig. 8c) then the resulting number is much larger.

Another situation for which we present only a small number of replicates is in cases where we repeated experiments that had previously been done with a strong promoter (e22c-Gal4) and had shown no effect (Supplementary Fig. 8), and we needed to use a weaker promoter (A58-Gal4) for the same construct as a control for other parts of the experiments (e.g. Fig. 6). Since the strong promoter had no effect, it seemed unnecessary to have extensive replicates for the weaker one, and we made do with a few cases as negative controls.

Finally, it is worth recalling that each wounding experiment requires long-term imaging, and microscope time is a limiting resource for these studies.

Reviewer #2

This elegant study sets up the *Drosophila* larval epithelium as a model system to study the molecular requirements for efficient wound healing. This allows powerful *Drosophila* genetics to be applied to a medically relevant question, addressing issues relating to signaling, cell biology, cytoskeletal rearrangements, etc.

This study looks at the role of the insulin and TOR signaling pathways in wound closure. The finding that these signaling pathways are required for wound healing is not unexpected. Surprising, instead, is the reason why. A priori, one would expect that insulin and TOR signaling are required acutely after wounding to promote cell growth. Instead, this study finds that the defective wound healing due to reduced insulin signaling can be rescued by removing FOXO. Since FOXO is a transcription factor, its effects are unlikely to take place within minutes - the timeframe of wound healing. Hence, it is likely that levels of insulin and FOXO signaling prior to the wounding predispose the cells for efficient or inefficient healing by modulating their transcriptional program. This is interesting and unexpected.

The data in this study are solid (and visually beautiful). The conclusions are

well founded. This study is likely to be the first in a series of studies exploiting this system to study wound healing.

I have one main concern, which is that some of the results and implications are not presented or discussed clearly enough, leading to confusion or perhaps incorrect conclusions by the reader:

1.) Figure 2 shows nicely that within minutes after wounding, PIP3 levels increase and FOXO becomes cytoplasmic. However, since loss of FOXO rescues the defective wound healing caused by reduced insulin signaling (Figure 8), and since the transcriptional effects of FOXO are unlikely to translate into changes in cell behavior within 30 minutes, the authors conclude that the effects of FOXO are probably due to the chronic reduction of FOXO activity prior to the wounding experiment, and not to the acute inactivation of FOXO upon wounding. This also means, however, that it is unclear whether the acute increase in PIP3 levels shown in Figure 2 have any physiological relevance in this healing context. This should be made clear and explicit in the presentation of the study.

Response:

Indeed, we make no claims about the relevance of the raised PIP3 level at the time of wounding. We just used them as a marker, and they do depend on intact Insulin receptor signalling. However, as with some of the other effects, we do not know whether this requirement of IIS for PIP3 accumulation is at the time of wounding, or throughout the preceding stages of development during which the cells are enabled to properly respond to wounding. We have made this more explicit in the text.

2.) Usually, increased autophagy correlates with (or is caused by) REDUCED IIS and TOR signaling, not INCREASED signaling. Hence, the elevated autophagy seen in Fig 2f clashes, or at least does not easily fit, with the elevated IIS signaling shown in Fig 2a.

Response:

This does at first appear as a conundrum, but we think that it is a question of timing. We would suggest that is not the early IIS (which takes FOXO out of the nucleus) that activates autophagy, but rather the FOXO that accumulates at very high levels in the nuclei after the initial response to wounding (which may be indicative of reduced IIS at this stage).

Even though the elevated autophagy is seen at 126 min after wounding, when PIP3 levels are returning to baseline and FOXO is becoming again more nuclear, there is nonetheless no indication (or data that I could see) suggesting that IIS or TOR signaling at this point have dropped BELOW the starting baseline levels, which could explain the autophagy induction.

Response:

True, we have no data on reduced IIS at this stage, and the interpretation is conjecture, justified exclusively by the high levels of FOXO in the nuclei at late stages of wound healing (which, at least in the case of the overexpressed FOXO, are in fact higher than at any of the earlier time points, hinting at reduced IIS). We have phrased this more clearly, and have also added question marks in our summary

diagram (Supplementary Fig. 13) for those connections for which we have no evidence. The new data from Brainbow clonal analyses, which show that the cells neighbouring the wound indeed ingest debris from the dead cell (New Fig. 3b) suggest a possible role for autophagy in digesting the debris.

In the current phrasing, the authors try to gloss over this issue. This should be made more clear/explicit. It is very possible that the autophagy induction in this context is mediated by an inflammatory response or ROS, stimulating clean-up of debris, rather than anything having to do with IIS signaling, so unless there is evidence for reduced IIS signaling at this phase of the healing process, presentation of the autophagy data should be separated from the IIS data in the manuscript, both logically and structurally.

Response:

We have added notes to this effect, and we have deleted from the discussion the over-speculative comments on the role of autophagy.

Minor Issues:

1. It is unlikely that localized production and/or release of dILPs at the site of wounding is causing the transient increase in PIP3 levels shown in Fig 2a.

Response:

We agree, and we did not intend to imply this. dILPs, just like InR signalling most likely are involved in the period during which the larva develops to set up the physiological state of the cells that allows them to respond efficiently.

(It would have to be very fast, and act very locally. It's possible, but maybe not). Alternatively, I believe Src can activate PI3K cell autonomously? Could it be that activation of Src in the cells adjacent to the wound is leading to the increased PIP3 levels? If so, perhaps this possibility should be presented in the text.

Otherwise, by presenting the dILP loss-of-function data in the following Figure 3, and by calling the signaling pathway "insulin/IGF signaling" gives the impression to the reader that the dILPs are likely playing a role in this context. Alternatively, is there reason to believe this effect is indeed mediated by dILP secretion?

Response:

We have no evidence for the acute role of dILPs themselves, apart from the role of the InR. As for the trigger for PIP3 synthesis, Src is a potential candidate – but there are probably others. For one, we have now included results from experiments that reveal a strong pulse of Ca⁺⁺ signalling within seconds of wounding (New Fig. 1f) (as has also been reported in other systems). So there is likely to be a parallel signal that leads to other, direct activities. Nevertheless, the finding remains that in a situation of reduced InR activity, the PIP3 response is also reduced. Again, this has now been phrased more carefully in the text.

2. Fig 2b: It would be helpful for the reader to see an image at lower magnification at 8m post wounding to show more clearly that indeed FOXO

shuttles from nucleus to cytoplasm in the cells facing the wound, but not in cells further away, as stated in the text.

Response:

We have included these images (New Fig. 2g,h and New Fig. 6c).

3. Fig 8a - is reduced accumulation of the PIP3 sensor upon InR[DN] expression due to reduced expression of the reporter? Or can this be excluded (e.g. with a GFP western from larval body walls?)

Response:

The sensor may indeed be expressed at lower levels, but what we are measuring here are the relative amounts of GFP associated with the cable versus levels in the cytoplasm, so the absolute starting point shouldn't matter too much. At the same time, GFP-levels are also reduced after FOXO overexpression, but nevertheless, PIP3 accumulates at similar levels as in the controls (but not similar to InR^{DN}) and the dynamics of actin cable assembly and contraction are similar to those in cells expressing InR^{DN} (but not as in controls).

4. Fig 9a-c would need just A58>InR[DN] or A58>foxo as control genotypes to show rescue.

Response:

The controls were in the previous figures. These experiments were done in parallel, but the results split into two figures for easier presentation. We have now stated this in the figure legend.

Reviewers' Comments:

Reviewer #1 (Remarks to the Author)

We think the new, larger wounds considerably add to the revised MS and while our concerns regarding significance of some of the data because of low n's remain, the authors explain how difficult it is to collect each movie and, given the number of complementary approaches/experiments, we are sufficiently convinced that these are robust and interesting findings.

Reviewer #2 (Remarks to the Author)

The authors have addressed the concerns raised in my original review.